# Novel sACE2-Anti-CD16VHH Fusion Protein Surreptitiously Inhibits SARS-CoV-2 Variant Spike Proteins and Macrophage Cytokines, and Activates Natural Killer Cell Cytotoxicity

**DOI:** 10.3390/vaccines13020199

**Published:** 2025-02-17

**Authors:** Abdolkarim Sheikhi, Leili Baghaie, Fatemeh Rahbarizadeh, Pooria Safarzadeh Kozani, Cobra Moradian, Mohammadreza Davidi, Narges Baharifar, Golnaz Kaboli, Mehdi Sheikhi, Yunfan Li, Mohammadamin Meghdadi, Abdulrahman M. Yaish, Aiden H. Yu, William W. Harless, Myron R. Szewczuk

**Affiliations:** 1Department of Biomedical & Molecular Sciences, Queen’s University, Kingston, ON K7L 3N6, Canada; 16lbn1@queensu.ca; 2Department of Immunology, School of Medicine, Dezful University of Medical Sciences, Dezful 64616-43993, Iran; baharinarges65@gmail.com (N.B.); golnaz.kaboli1375.gk@gmail.com (G.K.); 3Department of Medical Biotechnology, Faculty of Medical Sciences, Tarbiat Modares University, Tehran 14115-331, Iran; rahbarif@modares.ac.ir (F.R.); pooriasafarzadeh@modares.ac.ir (P.S.K.); s.moradian7@gmail.com (C.M.); 4Research and Development Center of Biotechnology, Tarbiat Modares University, Tehran 14115-331, Iran; 5Faculty of Medicine, Kazeroon Azad University, Kazeroon 14778-93855, Iran; moohammadreza.davidi@gmail.com (M.D.); mehsha_2012@hotmail.com (M.S.); 6Faculty of Arts and Science, Queen’s University, Kingston, ON K7L 3N9, Canada; 18yl210@queensu.ca; 7Faculty of Health Sciences, Queen’s University, Kingston, ON K7L 3N9, Canada; 21mame@queensu.ca (M.M.); a.yaish@queensu.ca (A.M.Y.); aiden.yu@queensu.ca (A.H.Y.); 8ENCYT Technologies Inc., Membertou, NS B1S 0H1, Canada; wharless@encyt.net

**Keywords:** COVID-19, sACE2-AntiCD16VHH, SARS-CoV-2, immunotherapy

## Abstract

**Background****/Objectives**: The SARS-CoV-2’s high mutations and replication rates contribute to its high infectivity and resistance to current vaccinations and treatments. The primary cause of resistance to most current treatments aligns within the coding regions for the spike S protein of SARS-CoV-2 that has mutated. As a potential novel immunotherapy, we generated a novel fusion protein composed of a soluble ACE2 (sACE2) linked to llama-derived anti-CD16 that targets different variants of spike proteins and enhances natural killer cells to target infected cells. **Methods**: Here, we generated a novel sACE2-AntiCD16VHH fusion protein using a Gly4Ser linker, synthesized and cloned into the pLVX-EF1alpha-IRES-Puro vector, and further expressed in ExpiCHO-S cells and purified using Ni^+^NTA chromatography. **Results**: The fusion protein significantly blocked SARS-CoV-2 alpha, beta, delta, gamma, and omicron S-proteins binding and activating angiotensin-converting enzyme receptor-2 (ACE2) on ACE2-expressing RAW-Blue macrophage cells and the secretion of several key inflammatory cytokines, G-CSF, MIP-1A, and MCP-1, implicated in the cytokine release storm (CRS). The sACE2-Anti-CD16VHH fusion protein also bridged NK cells to ACE2-expressing human lung carcinoma A549 cells and significantly activated NK-dependent cytotoxicity. **Conclusions**: The findings show that a VHH directed against CD16 could be an excellent candidate to be linked to soluble ACE2 to generate a bi-specific molecule (sACE2-AntiCD16VHH) suitable for bridging effector cells and infected target cells to inhibit SARS-CoV-2 variant spike proteins binding to the ACE2 receptor in the RAW-Blue cell line and pro-inflammatory cytokines and to activate natural killer cell cytotoxicity.

## 1. Introduction

SARS-CoV-S and SARS-CoV-2-S are coronaviruses that have lipid bilayers shaped by structural proteins, such as the spike S proteins and envelope proteins, embedded in the viral envelope. The heavily glycosylated spike S protein forms homotrimeric spike structures, which facilitate the viral infectivity of host cells [1]. Several cellular molecules have been described as the receptors for coronaviruses. The angiotensin-converting enzyme 2 (ACE2) is the main receptor of SARS-CoV-S and SARS-CoV-2-S spike S proteins. The ACE2 regulates the fluids, blood pressure, and electrolyte balance of the renin–angiotensin system (RAS). ACE2 regulates systemic vascular resistance [2,3,4,5]. SARS-CoV-2 can activate the pulmonary RAS in the lungs, causing lung injury through increased vascular permeability and alterations of alveolar epithelial cells [6,7]. ACE2 also hydrolyses angiotensin I (Ang I) to produce Ang1–9 and Ang II in generating Ang-(1–7), which binds to the MAS G-protein coupled receptor [8,9]. Agonist binding MAS can antagonize several Ang II-mediated cellular effects.

The human recombinant soluble ACE2 (sACE2) has been recently reported to inhibit the growth of SARS-CoV-2 in the Vero cell line [10]. This inhibition mechanism involves the competitive interaction of SARS-CoV-2 with ACE2 and hrsACE2. High Ang II levels can increase vascular permeability/pulmonary edema [11,12]. In preclinical mice models of acute respiratory distress syndrome disease, the knockdown of ACE2 produced severe symptoms, but the increased expression of ACE2 had protective effects [13]. SARS-CoV replication and its spike S protein can decrease the expression of ACE2 but not ACE in mice infected with the virus [14,15,16,17,18,19]. Indeed, the balance between Ang II/Ang (1–7) and ACE/ACE2 in the physiological stage can be interrupted by SARS-CoV infection, contributing to the pathogenic role in lung injury [3,20]. Interestingly, the SARS-CoV spike S protein has been shown to induce a significant increase in Ang II in the lung and aggravated acid-induced acute lung injury in mice [14]. Severe lung failure from high levels of Ang II inhibiting ACE2 may be a possible mechanism of SARS-CoV infection. SARS-CoV-2 and SARS-CoV might involve similar pathogenic mechanisms of disease progression [21]. The compensation of ACE2 in balance with ACE function may be a therapeutic target of the virus-induced severe lung injury [22].

The receptor-binding motif (RBM) of the SARS-CoV-2 S protein has conserved amino acid residues essential for the binding of S protein to ACE2. However, these S protein residues of the SARS-CoV virus from bats are absent and were not observed for ACE2 cellular entry [23,24,25]. The cellular entry of SARS-CoV-S and SARS-CoV-2-S viral proteins was blocked with anti-human ACE2 serum but not with the MERS-S virus protein into BHK-21 cells. The SARS-CoV-2 virus could infect high-efficiency ACE2-transfected BHK-21 cells but not the parental BHK-21 cells. These results suggest that both SARS-CoV-2-S and SARS-CoV-S use ACE2 for infectivity, even though the SARS-CoV-2-S spike protein has a higher affinity to ACE2 than the SARS-CoV-S protein. Recent reports have provided a possible mechanism of action involving the binding of SARS-CoV-S to ACE2 by inducing ACE2 shedding from the cells with higher efficiency than that of the NL63-S protein [15,26,27]. The lower affinity of NL63 for human ACE2 contributes to the milder pathogenicity of the NL63 virus (HCoV-NL63) than the SARS virus.

Furthermore, because the SARS-CoV-2-S protein has a higher affinity to ACE2 than SARS-CoV-S, unlike SARS-CoV, live SARS-CoV-2-infected cells form syncytium, suggesting that SARS-CoV-2 mainly utilizes the plasma membrane fusion pathway to enter and replicate inside cells [27,28]. The virus dangerously moves from cell to cell without the need to bud out because the S-protein on infected cells adheres to ACE2 on healthy neighbor cells, resulting in rapid lung infection. This major problem causes the death of patients, which makes us realize an urgent need for a drug before reaching this stage.

For the treatment of COVID-19, although the neutralization of the virus before entry could be a good approach, it should be accompanied by blocking the virus replication into the infected cells. To neutralize the virus and, at the same time, prevent virus replication into infected cells, we generated the sACE2-AntiCD16VHH fusion protein, as previously described [29]. The sACE2-AntiCD16VHH consists of the soluble ACE2 and the variable domain of the anti-human CD16 nanobody (VHH) obtained from a llama, joined together by a linker. We showed that the sACE2-AntiCD16VHH blocks different variants of SARS-CoV-2 spike molecules to bind to ACE-2 on the RAW-Blue macrophages. It also inhibited the secretion of pro-inflammatory cytokines from the SARS-CoV-2 S protein-stimulated RAW-Blue macrophages and mediated ADCC by NK cells against SARS-CoV-2 S protein-pretreated ACE2+ A549 target cells.

## 2. Materials and Methods

### 2.1. In Silico Study of the Soluble ACE2-AntiCD16VHH Construct

The detailed workflow of the whole silico design experiment is depicted in Figure 1.

### 2.2. Sequences and 3D Structures

Protein Data Bank (PDB; http://www.rcsb.org/pdb/home/home.do, accessed on 20 December 2021) was utilized for obtaining the 3D structures of FcγRIII (CD16) and the SARS-CoV-2 Spike protein. The National Center for Biotechnology Information (NCBI) Gene portal was searched and used for the protein sequence of ACE2. A literature search defined the amino acid residues of its soluble form [30,31,32]. A separate literature search was carried out with the aim of finding potential liner peptides [33,34,35,36,37,38]. Since the CD16-specific VHH was a novel single-domain antibody fragment previously isolated from an immune VHH library, various in silico techniques used for the engineering of VHHs were taken into consideration [39,40]. Also, whether the VHH should be linked to the soluble ACE2 protein through its N-terminus or C-terminus was assessed in in silico experiments.

#### 2.2.1. 3D Structure Prediction, Energy Minimization, and Flexibility Evaluation

Well-known protein structure prediction servers were utilized for predicting the 3D structures of the designed fusion proteins, including Robetta (https://robetta.bakerlab.org/), GalaxyWEB server (http://galaxy.seoklab.org/), I-TASSER (https://zhanggroup.org/I-TASSER/, accessed on 20 December 2021), and LOMETS (https://zhanggroup.org/LOMETS/, accessed on 20 December 2021). After the conformational structure of each of the designed fusion proteins was predicted, several tools were employed to assess the quality of the predicted models in an attempt to select the best server that provides 3D models with the utmost quality for further experiments. The servers used for structural quality assessments included QMEAN (https://swissmodel.expasy.org/qmean/, accessed on 20 December 2021), ProSA (https://prosa.services.came.sbg.ac.at/prosa.php, accessed on 20 December 2021), and MolProbity (http://molprobity.biochem.duke.edu/), which provided vast protein geometry analyses to help select the most qualified predicted models. Following the selection of the most qualified 3D model, an energy minimization process was taken into account. Briefly, the selected 3D model was further ameliorated by relieving severe clashes that might have occurred during structure prediction. To this aim, the ModRefiner server (http://zhanglab.ccmb.med.umich.edu/ModRefiner/, accessed on 20 December 2021) was employed to minimize the desired predicted structure energetically; therefore, future experiments would be carried out with more precision in terms of energetic stability. Additionally, UCSF Chimera software (version 1.17.3; San Francisco, CA, USA) was utilized to measure the average distance between the atoms of any superimposed proteins, which is known as root-mean-square deviation (RMSD), while PyMOL software (version 2.3.2; Schrödinger, LLC, New York, NY, USA) was employed for high-quality visualizations. The most qualified fusion construct was ultimately evaluated in terms of flexibility using the server CABS-flex 2.0 (http://biocomp.chem.uw.edu.pl/CABSflex2, accessed on 20 December 2021).

#### 2.2.2. Characterization

The physicochemical properties and the melting temperature (Tm) of the most qualified fusion protein were determined utilizing the ProtParam server (http://web.expasy.org/protparam/, accessed on 20 December 2021) and the Tm Predictor server (http://tm.life.nthu.edu.tw/index.htm, accessed on 20 December 2021), respectively. The antigenicity profile of the fusion protein was analyzed using VaxiJen (http://www.ddg-pharmfac.net/vaxijen/VaxiJen/VaxiJen.html, accessed on 20 December 2021). Additionally, the potential B-cell epitopes of the desired fusion protein were examined to validate its potential clinical applicability further using the Bepipred Linear Epitope Prediction tool (http://tools.iedb.org/bcell/, accessed on 20 December 2021) with the threshold set at 0.0350. The ccSol omics server (http://service.tartaglialab.com/new_submission/ccsol_omics, accessed on 20 December 2021) was used for the prediction of the solubility propensity of the desired fusion protein, while its solubility profile was predicted using the Aggrescan3D server (https://biocomp.chem.uw.edu.pl/A3D2, accessed on 20 December 2021).

#### 2.2.3. Binding Capacity Assessment

The final phase of the in silico experiments entailed an assessment of the binding capacity of the desired fusion construct to CD16 (via the VHH) and SARS-CoV-2 Spike (via ACE2), as the ClusPro (https://cluspro.bu.edu/) server was utilized for carrying out this step.

### 2.3. Gene Construct

sACE2-AntiCD16VHH (with the addition of a C-terminal hexaHis tag) fragments were synthesized (Genscript Biotechnology, Nanjing, China) and delivered on the pUC57 vector. The pLVX-EF1alpha-IRES-Puro vector (Takara, Brussels, Belgium) and sACE2-AntiCD16VHH- hexaHis fragment were digested by *Eco*RI/*Bam*HI (Fermentase, Sankt Leon-Rot, Germany). Then, the ACE2-AntiCD16VHH gene fragment in the gel was isolated and further purified using a gel extraction kit (Roche, Brussels, Belgium). Also, the enzymatic digestion product of the pLVX-EF1alpha-IRES-Puro vector was purified using the Clean-up kit (Roche). These two fragments were then used for the ligation reaction to make the pLVX/sACE2-AntiCD16VHH-hexaHis construct. For enzymatic ligation, the molar ratio of the fragment to the vector was chosen to be 3:1, and the ligation reaction was placed at 16 °C for 16 h. The ligation product was used to transform susceptible *Stbl4* bacteria using a chemical method (calcium chloride). Then, the construct was purified using an endotoxin-free plasmid DNA purification kit (Macherey-Nagel, Future Lab Innovations, Brussels, Belgium) and confirmed by colony PCR, digestion, and sequencing. In order to perform colony PCR and sequencing, two forward primers and one reverse primer were designed and synthesized (Neday Fan Company, Tehran, Iran), and their sequences are given below.

Forward1: AGCCTCAGACAGTG

Forward2: ATCCAGAACCTGAC

Reverse: ACGGCAATATGGTG

### 2.4. Cell Lines

RAW-Blue™ cells are mouse macrophage reporter cell lines (InvivoGen, San Diego, CA, USA) that are derived from RAW 264.7 macrophages. They are grown in a culture medium containing Zeocin as the selectable marker, supplemented with 10% fetal bovine serum (FBS) and 0.1% plasmocin.

RAW-Blue™ cells are resistant to Zeocin™ and G418 in the conditioned medium. ExpiCHO-S cells (simAbs Company, Diepenbeek, Belgium), a CHO-S cell line with a high-efficiency transfection efficiency, were used for transfection with the pLVX/sACE2-AntiCD16VHH-hexaHis plasmid using the ExpiFectamine CHO (Thermo Fisher Scientific, Waltham, MA USA) transfection kit. The A549 cell line was cultured in RPMI (Gibco, Thermo Fisher Scientific, Waltham, MA USA) medium supplemented with 10% FBS and 100 U/mL penicillin, 100 μg/mL streptomycin, and 0.3 mg/mL Glutamine. A549 cell lines were used as target cells for the NK cytotoxicity assay.

### 2.5. Ligands and Reagents

Lipopolysaccharide (LPS, BioVision, Toronto, ON, Canada) at 50 μg/mL was used in the experiments as a positive control. 2-(4-Methylumbelliferyl)-α-D-N-acetylneuraminic acid (98% pure 4-MUNANA; Biosynth International Inc., Itasca, IL, USA), a sialidase enzyme substrate at a pre-determined concentration of 0.318 mM diluted in TBS, was used in the live cell sialidase assay experiments.

Recombinant spike His-tag proteins (R&D Systems, a Biotechne brand) were used with the following samples: Alpha variant (UK, SARS-CoV-2 B.1.1.7 N501Y Spike His-tag Protein, CF, Catalog # 10748-CV-100), Beta variant (S. Africa, SARS-CoV-2 B.1.351 Spike RBD His Protein, CF, Catalog # 10735-CV-100), Delta variant (India, SARS-CoV-2 B.1.617.2 Spike GCN4-IZ Protein, CF, Catalog # 10878-CV-100), Gamma variant (Brazil, SARS-CoV-2 P.1 Spike GCN4-IZ His-tag Protein, CF, Catalog # 10795-CV-100), Kappa variant (India, SARS-CoV-2 B.1.617.1 Spike GCN4-IZ Protein, CF, Catalog # 10861-CV-100), BA.2 variant (omicron, SARS-CoV-2 BA.2 Spike (GCN4-IZ) His Protein, CF, Catalog # 11109-CV-100), BA.4 variants (omicron, SARS-CoV-2 BA.4 Spike His-tag Protein, CF, Catalog # 11232-CV-100), and Omicron variant (recombinant SARS-CoV-2 B.1.1.529 spike S1 protein, His-tag, Catalog # 40589-V49H6-B, Sino Biological US Inc., 10101 Southwest Freeway, Suite 100, Houston, TX 77074, USA).

### 2.6. Production and Purification of sACE2-anti CD16VHH Fusion Protein

To generate sACE2-antiCD16VHH-hexaHis fusion protein, ExpiCHO-S cells (simAbs company, Belgium) were transfected with the pLVX/sACE2-AntiCD16VHH-hexaHis plasmid using the ExpiFectamine CHO transfection kit. The clarified harvest was loaded onto Ni^+^NTA chromatography columns, and the fusion protein was eluted using a 300 mM imidazole elution buffer. Eluted fractions were buffer-exchanged to 1× PBS using a spin column device. After buffer exchange, the solution was sterile-filtered through a 0.2 µm filter and kept at 4 °C or −20 °C for long-term storage.

### 2.7. Sialidase Activity Assay

RAW-Blue macrophage cells were grown on circular 12 mm glass slides for 24 h at 37 °C in a humidified incubator. Removing the media, 0.318 mM 4-MUNANA substrate in Tris-buffered saline, pH 7.4, was added to cells alone (background, Bkg) or with indicated S-proteins, 20 μg/mL or 50 ug/mL LPS. With 100 μg/mL of sACE2 fusion protein in combination with the indicated S-proteins, the sialidase expression was markedly reduced. Fluorescent images were taken using epi-fluorescent microscopy (Zeiss Imager M2, 20× objective, Toronto, ON, Canada) at 2 min after adding substrate. A scatter plot of data point visualization using dots was used to represent the fluorescence values (n = 50) obtained from one representative experiment of 3 separate experiments with similar results. The mean fluorescence intensity of 50 different points surrounding the cell was quantified using Image J software, 1.5g, Java 1.8.0_345 (64-bit). The relative fluorescence values of each group were compared to the indicated group by ANOVA using a multiple comparisons test with uncorrected Fisher’s LSD at 95% confidence. The statistical significance is indicated with appropriate asterisks.

### 2.8. MILLIPLEX^®^ Luminex^®^ 200™ xMAP^®^ Flow Cytometry-Based Instrumentation

The MILLIPLEX^®^ Luminex^®^ 200™ xMAP^®^ flow cytometry-based instrumentation (MilliporeSigma, St. Louis, MO, USA) protocol is reported and described in detail [41]. Seventeen tissue culture supernatants were prepared and shipped on ice to Encyt Technologies Inc., where they were frozen at −80 °C upon receipt. The samples on the day of testing were thawed, vortexed, and then centrifuged at 14,000 rpm for 1 min to remove the particles. For testing, the samples were prepared using the procedure in the MILLIPLEX^®^ Mouse Cytokine/Chemokine Magnetic Panel User Guide for the selected cytokines (Kit Catalog# Hcytomag-70K, Lot#3893941, Millipore Sigma, St. Louis, MO, USA.). The components in the kit were stored at 2–8 °C until the day of testing. The cytokines tested in this kit are G-CSF, MIP-1A, MCP-1, and IL-6.

### 2.9. Peripheral Blood Mononuclear Cells and Natural Killer Cell Isolation

Blood samples were obtained from healthy volunteers. Natural killer (NK) cells were directly isolated from whole blood through immunomagnetic isolation using the EasySep™ Direct Human NK Cell Isolation Kit (STEMCELL Technologies, Toronto, ON, Canada), following the manufacturer’s instructions. NK purity and yield were checked in our previous study [35]. Briefly, the whole blood sample was added to the required tube. The Isolation Cocktail was mixed with the sample and incubated at RT for 5 min. The RapidSpheres™ was mixed into the sample. With RPMI, the sample volume was topped up to 2.5 mL. The sample was mixed by gently pipetting. The tube was placed into the magnet container and incubated at RT for 3 min. The magnet was removed, and the enriched NK cell suspension was poured into a new tube.

### 2.10. NK Cytotoxicity Assay

The immunomagnetically separated human NK cells and SARS-CoV-2 S protein (UK variant) pretreated or untreated (control) ACE-2-expressing A549 target cells were co-cultured with sACE2-Anti-CD16VHH fusion protein (3.4 µg/mL). The cells were co-cultured in an NK-to-A549 cell ratio of 1:1 (10^5^ cells/well), with a total volume of 100 μL RPMI in a 96-well plate, for 4 h. Cytotoxicity was determined using the AlamarBlue reagent. Ten µL of AlamarBlue was added to every 100 µL of supernatant and incubated for 4 h at 37 °C and 5% CO_2_. The absorbance was recorded using (ex 560 nm; em 590 nm). The % cytotoxicity was calculated using the following formulas:% cytotoxicity = [OD of S-pretreated target cells + VHH + NK/(OD of S-pretreated Target cells + NK)] × 100% cytotoxicity = [(OD of untreated target cells + VHH + NK/(OD of Un-treated Target cells + NK)] × 100

### 2.11. Statistical Analysis of Data

Comparisons between two groups from three independent experiments were made by one-way analysis of variance (ANOVA) at 95% confidence using the uncorrected Fisher’s LSD multiple comparisons test with 95% confidence with asterisks denoting statistical significance. The data are presented as the mean ± the standard error of the mean (SEM) from three repeats for each experiment performed in triplicate.

## 3. Results

### 3.1. Results of In Silico Design Experiments

#### 3.1.1. Sequences and 3D Structures

The structure of the SARS-CoV-2 crystallized spike glycoprotein under the PDB ID of 6VXX was selected for further use in the docking process since it was the conformationally intact cryo-EM structure of the protein as the wild-type SARS-CoV-2 viral particles would have decorated on their surface [42]. Moreover, 1E4J was considered as the crystal structure of the soluble FcγRIII to be used in the docking process [43]. The soluble form of ACE2 protein was selected from the 18th to 615th amino acid of the whole protein since this fragment has been known to retain its enzymatic capability and binding site for the SARS-CoV-2 Spike protein [44,45,46]. Three versions of the flexible linker peptide GGGGS (GGGGS1, GGGGS2, and GGGGS3) alongside the rigid linker “PAPAP” and the helix-forming linker “AEAAAKEAAAKA” were used in the construction of the fusion protein. On this basis, the following constructs were designed (N-terminus to C-terminus, respectively): VHH-GGGGS1-ACE2, VHH-GGGGS2-ACE2, VHH-GGGGS3-ACE2, ACE2-GGGGS1-VHH, ACE2-GGGGS2-VHH, ACE2-GGGGS3-VHH, VHH-PAPAP-ACE2, ACE2-PAPAP-VHH, VHH-AEAAAKEAAAKA-ACE2, and ACE2-AEAAAKEAAAKA-VHH.

#### 3.1.2. 3D Structure Prediction, Energy Minimization, and Flexibility Evaluation

The model with the highest quality out of the predicted 3D models of each of the four servers was used for further evaluation in terms of structural validity, as these predicted structures were put to the test to evaluate their structural quality. Table 1 represents the predicted model data of the QMEAN score value, which predicts the structure, the Z-score obtained from ProSA, and the results for the Ramachandran plot evaluations of each model. According to these findings, Robetta was found to provide predicted 3D models with the highest quality. Figure 2a–c presents the 3D structure of the CD16-specific VHH, soluble ACE2, and the used linker peptides, respectively. Moreover, VHH-GGGGS_3_-ACE2, VHH-PAPAP-ACE2, and VHH-AEAAAKEAAAKA-ACE2 were selected as the top three fusion constructs (Figure 2d–f).

Furthermore, to determine whether the fusion of VHH and ACE2 via the GGGGS_3_, PAPAP, and AEAAAKEAAAKA linkers could negatively impact their conformational structure, the RMSD values of the fusion constructs as aligned with the VHH and ACE2 were determined (Figure 2g–i). Briefly, the RMSD between the VHH and VHH-GGGGS_3_-ACE2, VHH-PAPAP-ACE2, and VHH-AEAAAKEAAAKA-ACE2 were calculated as 1.224, 1.546, and 1.216 Å, respectively. Moreover, the alignment of ACE2 with VHH-GGGGS_3_-ACE2, VHH-PAPAP-ACE2, and VHH-AEAAAKEAAAKA-ACE2 demonstrated RMSD values of 1.260, 1.364, and 1.121 Å, respectively. These findings indicated that the addition of the linker does not impinge on the native conformational structure of either protein in the format of the fusion construct. Ultimately, we decided that the fusion construct with the GGGGS_3_ was the more qualified compared with the other two due to the flexibility of the linker; therefore, the rest of the experiments were conducted with VHH-GGGGS_3_-ACE2.

The energy minimization processing results exhibited very slight changes to the structure of VHH-GGGGS^3^-ACE2 before and after the process (Figure 3a). The estimated RMSD was calculated to be 0.472 Å (with a TM-score of 0.9976 to the initial model), indicating that minimal improvements were made to the structural accuracy of VHH-GGGGS^3^-ACE2. Moreover, the evaluation of the flexibility of VHH-GGGGS^3^-ACE2 demonstrated sharp spikes in RMSF values (between 8 and 10 Å) between residues 123 and 137, indicating the flexibility of the linker peptide as desired (Figure 3b,c).

#### 3.1.3. Characterization

The estimated molecular weight and theoretical pI of VHH-GGGGS^3^-ACE2 were calculated to be 83,431.80 Da and 5.13, respectively. The chemical formula of the construct is determined as C_3741_H_5634_N_986_O_1122_S_33_. Moreover, the half-life of VHH-GGGGS^3^-ACE2 was estimated to be 1 h, 30 min, and more than 10 h in vitro in yeast and *Escherichia coli* (*E. coli*)*,* respectively. The grand average of hydropathicity and the aliphatic index were calculated to be 71.27 and −0.455, respectively. Moreover, Tm Predictor predicted the Tm of VHH-GGGGS^3^-ACE2 as higher than 65 °C.

Furthermore, according to the results of VaxiJen, the overall antigenicity index of VHH-GGGGS^3^-ACE2 was predicted as 0.3692, meaning that this protein is designated as a non-antigen. The Bepipred Linear Epitope Prediction tool for the immunological analyses gave VHH-GGGGS^3^-ACE2 an average score of 0.120 (with a minimum score of −0.002 and a maximum score of 2.588) with the threshold set at 0.350, indicating that VHH-GGGGS^3^-ACE2 is below the defined threshold (Figure 3d). The solubility probability of VHH-GGGGS^3^-ACE2 was predicted as 94%, meaning that this protein has a 94% probability of being soluble in *E. coli*. (Figure 3e). Based on the results of the Aggrescan3D server, VHH-GGGGS^3^-ACE2 was given an average score of -0.8386 (with a maximum and minimum score of 1.9407 and −4.1038, respectively), indicating that the fusion construct is highly soluble (Figure 3f).

#### 3.1.4. Binding Capacity Assessment

The PDB ID of 1E4J was selected as the 3D structure for the CD16 molecule, while 6VXX was selected as the crystal structure of the SARS-CoV-2 Spike protein. 1E4j is the crystal structure of the soluble human FcγRIII; therefore, it seemed fit for this experiment. The docking results demonstrated the binding capability of VHH-GGGGS_3_-ACE2 to CD16 via the CDR segments of the VHH (Figure 4a). This indicates that the fusion of the VHH to ACE2 does not negatively alter its native conformational structure nor result in the masking of the VHH CDRs. Moreover, the ability of the ACE2 segment of VHH-GGGGS_3_-ACE2 to bind SARS-CoV-2 Spike protein was also confirmed by the docking results (Figure 4b).

### 3.2. Subcloning of solubleACE2-AntiCD16VHH Fragment from Plasmid pUC57 to Viral Vector pLVX-EF1alpha-IRES-Puro

The enzymatic digestion of the pUC57/solubleACE2-AntiCD16VHH bi-specific molecule by *EcoRI* and *BamHI* (Fermentase) resulted in two bands with sizes of 2671 bp and 2315 bp. Then, the samples were electrophoresed on 1% *w*/*v* agarose gel. The 2315 bp band is the desired gene fragment (Figure 5).

Then, the 2315 bp band related to the sACE2-AntiCD16VHH gene fragment was cut from the gel and purified using the gel extraction kit (Roche) and electrophoresed on 1% *w*/*v* agarose gel (Figure 6a). The enzymatic digestion products of pLVX-EF1alpha-IRES-Puro vector with *EcoRI* and *BamHI* enzymes were purified using the Clean-up kit (Roche), and were electrophoresed on 1% *w*/*v* agarose gel (Figure 6b).

### 3.3. Confirmation of sACE2-AntiCD16VHH Cloning in pLVX-EF1alpha-IRES-Puro Vector by Colony PCR

After transformation, colony PCR was performed using forward 1 and reverse primers for eight different colonies. These primers were attached to the positions on the pLVX plasmid just before and after the sACE2-AntiCD16VHH gene and amplified a fragment of 2486 bp. Finally, the colony corresponding to well number 4, which had the desired band, was selected as a positive colony and was considered for plasmid purification and further work (Figure 7).

### 3.4. Confirmation of sACE2-AntiCD16VHH Cloning in pLVX by Enzymatic Digestion

pLVX/sACE2-AntiCD16VHH was purified from the transformed bacterial colony and digested by *XhoI* enzyme (which has two cutting sites in this plasmid) and *HindIII* (which has six sites) and simultaneously digested with *EcoRI* and *BamHI*, that is, the same two enzymes with which the gene fragment was cloned into the plasmid. Then, the samples were electrophoresed on 1% *w*/*v* agarose gel (Figure 8).

### 3.5. Confirmation of sACE2-AntiCD16VHH Cloning in pLVX Vector by Sequencing

The pLVX/sACE2-AntiCD16VHH was sent to the Gen Fanavaran company for sequencing by the specific primers. The sequencing results were then aligned against the original sequence using MEGA 6.0. The result confirmed that the sequence of the construct was accurate and without any mutations (Figure 9, Figure 10 and Figure 11).

### 3.6. Production and Purification of an sACE2-anti CD16VHH Fusion Protein

To generate sACE2-anti CD16VHH-hexaHis fusion protein, ExpiCHO-S cells (simAbs company, Belgium) were transfected with the pLVX/sACE2-AntiCD16VHH-hexaHis plasmid using the ExpiFectamine CHO transfection kit. The clarified harvest was loaded onto Ni^+^NTA chromatography columns, and the fusion protein was eluted using a 300 mM imidazole elution buffer. Eluted fractions were buffer exchanged to 1× PBS using a spin column device. After buffer exchange, the solution was sterile-filtered through a 0.2 µm filter and kept at 4 °C or −20 °C for long-term storage.

### 3.7. sACE2-anti CD16VHH Fusion Protein Production

To generate the sACE2-anti CD16VHH fusion protein, the following protocol was followed using ExpiCHO-S cells.

#### 3.7.1. Cell Preparation

ExpiCHO-S cells (simAbs company, Belgium) were thawed directly into pre-warmed ExpiCHO Expression Medium and cultured at 37 °C with 8% CO_2_ in a humidified incubator. The cells in suspension culture were allowed to recover and grow to a density of 4–6 million viable cells per milliliter over 3–4 days post-thaw. Cell viability and density were monitored regularly, ensuring that viability remained above 95%.

#### 3.7.2. Subculturing and Transfection Preparation

On the day prior to transfection, the ExpiCHO-S cells at a final density of 3–4 million viable cells per milliliter were subcultured and allowed to grow overnight. On the day of transfection, the cell density and viability were re-examined, and the cells were diluted to a final density of 6 million viable cells per milliliter with fresh, pre-warmed ExpiCHO Expression Medium in 50 mL volumes within triplicate 250-milliliter shake flasks.

#### 3.7.3. Transfection

The pLVX/sACE2-AntiCD16VHH plasmid was transfected into the ExpiCHO-S cells using the ExpiFectamine CHO transfection kit. For the transfection, 3.2 microliters of ExpiFectamine CHO reagent and 0.8 micrograms of plasmid DNA per milliliter of culture were used. The reagents were prepared in cold conditions, and the DNA–reagent complexes were formed by gently mixing the components without vortexing or pipetting vigorously. The complexes were then added to the cell cultures, and the flasks were gently swirled to mix the cells and reagents evenly.

#### 3.7.4. Post-Transfection Culture

After transfection, the cells were incubated in the same ExpiCHO Expression Medium without the need for a media change, as this medium is specifically formulated to support transfection and protein expression. The cultures were maintained on an orbital shaker platform at 37 °C in a humidified incubator with 8% CO_2_. Protein expression was monitored, and the culture was harvested between 5 and 14 days post-transfection, depending on the optimal expression time for the sACE2-anti CD16VHH fusion protein. The fusion protein concentration was 643 µg/mL.

#### 3.7.5. sACE2-AntiCD16VHH Fusion Protein Purification

The purification of the sACE2-AntiCD16VHH protein, which has a molecular weight of approximately 83 kDa, was carried out using Ni-NTA chromatography spin columns following the protocol recommended by the supplier. Briefly, the cell lysate containing the 6xHis-tagged sACE2-AntiCD16VHH protein was prepared and adjusted to the appropriate buffer conditions. Up to 600 µL of the cell lysate was loaded onto the pre-equilibrated Ni-NTA spin column. The column was then centrifuged at 700× *g* for 2 min to facilitate the binding of the His-tagged protein to the Ni-NTA silica. The Ni-NTA spin column was washed with Wash Buffer (typically containing 20 mM sodium phosphate, 300 mM sodium chloride, and 20–50 mM imidazole) to remove non-specifically bound proteins. This step was repeated two to three times to ensure the removal of contaminants. The bound sACE2-AntiCD16VHH protein was eluted from the Ni-NTA resin using 300 mM imidazole elution buffer. By following these steps, the sACE2-AntiCD16VHH protein was effectively purified to a high level of purity, ready for further analysis or application.

#### 3.7.6. SDS-PAGE

The sACE2-AntiCD16VHH (MW: 83 kDa) was eluted from the Ni-NTA. The eluted protein was prepared for SDS-PAGE analysis by mixing it with a sample buffer and further boiling it for 5 min at 95 °C. Then, it was loaded onto a polyacrylamide 12% gel. The SDS-PAGE was run at a constant voltage of 100 V for 2 h. The gel was stained with Coomassie after the electrophoresis (Figure 12).

#### 3.7.7. Western Blot

Samples were taken from the transfected culture. These samples were centrifugated to separate the cell pellet from the harvest media. The obtained cell pellets were consequently lysed (incubation with lysis buffer followed by sonication), and cell supernatant was separated from cell debris after a second centrifugation step. The sACE2-AntiCD16VHH fusion protein with the His tag was produced and analyzed using a Western blot and anti-His antibody (Catalog #: MAB050, Bio-Techne, Toronto, ON, Canada) (Figure 13).

#### 3.7.8. Functional Assays

##### sACE2-AntiCD16VHH Fusion Protein Blocks SARS-CoV-2 Spike Protein S1-Activated Neu-1 Sialidase Activity in Murine Macrophages

Neu-1 was reported to regulate a conserved ACE2 signaling platform involved in receptor activation and the Toll-like receptors (TLRs) implicated in the cytokine release storm (CRS) [41]. Here, activated Neu-1 cleaves alpha-2,3- sialic acid residues, removing steric hindrance to both ACE2 and TLR dimerization. This Neu1 process was found to be critical to both viral attachment to the receptor and entry into the cell and TLR activation. As depicted in the infographic image in Figure 14A, by blocking Neu-1 with oseltamivir phosphate or aspirin, ACE2 receptor dimerization and internalization are inhibited, as well as TLR activation. Several key inflammatory molecules have been implicated in the CRS and death from ARDS [41]. To investigate whether the sACE2-AntiCD16VHH fusion protein can block the SARS-CoV-2 spike S proteins binding to ACE2, we used different variants, alpha, beta, kappa, gamma, omicron, BA-2, and BA-4, of SARS-CoV-2 spike S proteins with sACE2-AntiCD16VHH fusion protein or without as a control using RAW-Blue macrophages in a sialidase assay. The sACE2-AntiCD16VHH fusion protein, as shown in Figure 14B,C, significantly inhibited Neu-1 sialidase activity induced by S proteins from SARS-CoV-2 alpha (UK), beta, kappa, gamma, omicron, BA.4, and BA.5 in RAW-Blue cells.

##### Cytokine Assay

Cytokine storm syndrome can lead to ARDS and is a contributing cause of mortality in COVID-19. Here, we investigated a recombinant SARS-CoV-2 S protein (omicron and BA-2 variants) inducing cytokine and chemokine responses to stimulation of the mouse RAW-Blue macrophage cell line and the modulatory effect of sACE2-AntiCD16VHH fusion protein on distinct cytokines and chemokines secretion. MCP-1, G-CSF, and MIP1-α chemokines are involved in the increased recruitment of immune cells in activating macrophages, which were shown to be increased in patients with ARDS [47]. Here, we tested the effect of sACE2-AntiCD16VHH fusion protein treatment on GCSF, MIP1-α, and MCP-1 and production in response to different SARS-COV-2 omicron and BA-2 variants stimulation (Figure 15). sACE2-AntiCD16VHH fusion protein pretreatment significantly reduced G-CSF and MIP-1A chemokines secretion in response to omicron and BA.2 variants protein. Interestingly, the fusion protein pretreatment of the BA-2 sub-variant did not have a significant inhibitory effect on MCP-1 and IL-6 secretion compared to the BA-2 sub-variant alone but had a minor effect on omicron spike protein (Figure 15). Notably, these inhibitory effects significantly reduced G-CSF and MIP-1A by approximately 99% compared to the BA-2 sub-variant alone (Figure 15).

##### NK Activity Assay

Also, to show if the sACE2Anti-CD16VHH fusion protein bridges the NK cells to SARS-CoV-2-infected cells and triggers antibody-dependent cell cytotoxicity (ADCC), we co-cultured MACS separated human NK cells and ACE-2+ A549 target cells with or without (control) SARS-CoV-2 S protein for 4 h. The sACE2-AntiCD16VHH fusion protein and the NK cytotoxicity were measured by Alamar Blue colorimetric assay (Figure 16). There was a marked increase but not a significant percentage of NK cytotoxicity targeting ACE-2+ A549 target cells compared to the control without S protein.

## 4. Discussion

There are some virus neutralizations suggested before entry, such as targeting viral glycoprotein directly by neutralizing antibodies (such as antibodies produced in the community because of immunization). These neutralizing antibodies may not react to resistance to other mutants of the virus in the future. Targeting the viral receptor on the human cell surface is another strategy. The disadvantage of this technique could be that it disrupts the ACE2 function or stimulates the immune response against the ACE2. Soluble ACE2 (sACE2) may neutralize the binding of the S protein of SARS-CoV-2.

Soluble recombinant human ACE2 has been reported to inhibit the entry of SARS-CoV-2 into human cells [10]. Since SARS-CoV-2 has more affinity for ACE2 compared to SARS-CoV, it could be a promising strategy for the prevention and future treatment of COVID-19 [30,31].

Here, we focused on constructing a chimeric immunoadhesin molecule by attaching sACE2 to a human IgG Fc fragment. The Fc fragment of the chimeric molecule extends the half-life of sACE2 [48]. This ACE2-Fc construct, compared to neutralizing, not opsonizing antibodies, has the effector functions of the Fc fragment to recruit macrophages, natural killer, and dendritic cells via the FcγR receptor against the particles of the virus or infected cells. This may facilitate faster activation of immune response and eliminate the virus. Some evidence illustrates that engaging antibodies were more potent in eliminating SARS via activation of phagocytic cells than antibodies that neutralize the virus [49,50].

The Fc domain of IgG can bind to both the activating and inhibitory Fc receptors (FcγRIIB) that are expressed in B cells and myeloid cells [35,36]. Human IgG receptors (FcγR) have three classes: CD64 (FcγRI), CD32 (FcγRII), and CD16 (FcγRIII). CD16 (FcγRIII) and CD64 (FcγRI) are activating, and CD32 (FcγRII) isoforms are inhibitory receptors. The transmembrane isoform of CD16 has a low affinity for IgG and is expressed on NK cells, a small T lymphocyte subpopulation, as well as monocytes and macrophages. The activating FcγR receptor is involved in antibody-dependent cell-mediated cytotoxicity (ADCC), phagocytosis, endocytosis, and cytokine release. The chimeric ACE2-Fc molecule might not be effective against the virus due to the size of the Fc fragment and its low affinity for CD16. Here, we constructed a chimeric molecule with single-domain antibodies (sdAbs) and the variable domain of the camelid heavy-chain antibodies (VHH) that might be a proper strategy for the treatment of COVID-19 [37,51]. These small antibody domains may have a large number of properties, making them very attractive for antibody engineering. VHH domains exhibit affinities in the range of those of conventional monoclonal antibodies (mAbs) despite the reduced size of their antigen-binding surface [38].

Without requiring the use of an artificial linker peptide (as for ScFv) or of bi-cistronic constructs (as for Fab fragments), the single-domain nature of VHH can induce the amplification and subsequent cloning of the corresponding genes. The VHH format produces high amounts of VHH-based fusion molecules when expressed. The VHH fragments show exquisite refolding capabilities and physical stability [39]. Finally, genes encoding VHH show a large degree of homology with the IGVH3 family of human IGVH genes [40], conferring a low antigenicity in humans.

Altogether, these data show that a VHH directed against CD16 could be an excellent candidate to be linked to soluble ACE2 to generate a bi-specific molecule (sACE2-AntiCD16VHH) [42] suitable for bridging effector cells and infected target cells (Figure 17).

Therefore, based on data from this study, we generated an sACE2-antiCD16VHH fusion protein that has the potential to neutralize the virus and, at the same time, could prevent virus replication in cells.

We showed that the sACE2-antiCD16VHH fusion protein significantly blocks different SARS-CoV-2 spike molecules, including Alpha (UK, B.1.1.7), Beta (S. Africa, B.1.351), Delta (India, B.1.617.2), Gamma (Brazil, P.1), Kappa (India, B.1.617.1), BA.2 (omicron), and BA.4 variants (omicron) to bind to ACE-2 on the RAW-Blue macrophages. Also, we showed that the sACE2-antiCD16VHH fusion protein inhibits the secretion of inflammatory cytokines, including G-CSF, MIP-1A, and MCP-1, but not IL-6, from the RAW-Blue macrophages after stimulation by the Omicron and BA.2 S proteins. These data confirm the flexibility of the sACE2-antiCD16VHH fusion protein to neutralize different SARS-CoV-2 variants, which indicates that the therapeutic capability of this fusion protein is not lost with the change and mutation of the SARS-CoV-2 virus.

According to the ability of this fusion protein to prevent the binding of the virus spike to the ACE2, it can be concluded that this fusion protein could prevent the virus from disturbing the balance between ACE2/ACE and angiotensin II/I, thereby removing edema/pulmonary permeability.

On the other hand, we showed that the sACE2-AntiCD16VHH fusion protein markedly enhances cytotoxicity against SARS-CoV-2 S protein-pretreated ACE-2+ A549 target cells compared to untreated cells (Figure 16), but not significantly, due to variation in the NK cell’s cytotoxicity. This means that the fusion protein can provide a bridge from the NK cells to the infected target cells and facilitate the killing of infected cells. This function is a kind of ADCC, which finally would lead to preventing the spread of infection in the patient’s body. The results show potential promising therapeutic activation of NK cytotoxicity using the sACE2-AntiCD16VHH fusion protein. One limitation of our study is that we did not investigate if the sACE2-AntiCD16VHH can augment NK cytotoxicity against SARS-CoV-2-infected cells in vivo due to biohazard limitations on using virally infected cells in animals.

In contrast to Fc, the VHH does not activate a complement and has a high affinity for FcR, and, thus, it strongly activates ADCC and rapidly prevents the proliferation of the virus in infected cells. The small size of the VHH permits it to penetrate tissues rapidly. Although its size is smaller than sACE2-Fc, due to its high affinity to CD16, its half-life would not be less than that of SACE2-Fc.

ACE-2 expression is found in various parts of our body and is not limited to the epithelial cells of the respiratory tract, providing an easy entrance for the virus at different locations in the body. ACE-2 expression is also identified in the neurons of the spinal dorsal horn, a critical site for the manifestation of pain signals. In the spinal dorsal horn, the ACE/Ang II/AT1 receptor signaling pathway can facilitate pain transmission together with direct or indirect tissue damage induced by SARS-CoV-2 infection. The virus can invade the ACE-2-containing neurons and microglia in the spinal cord, causing a decrease in ACE-2^+^ cells with angiotensin II accumulation and angiotensin (1–7) reduction. This activation imbalance in ACE/ACE2 can ultimately induce pain from the spinal dorsal horn. ACE-2 on other cells in the body may be affected similarly, which is a possible cause of myalgia and widespread pain in COVID-19 patients. The ACE/Ang (1–7)/Mas receptor signaling pathway involved in the inhibition of the phosphorylation of the p38 mitogen-activated protein kinase may mitigate these pain signals [43,44].

In a study conducted in China, fecal samples collected from COVID-19 patients showed positive for viral RNA even after nasal samples were declared to be negative. These results indicated that the SARS-CoV-2 invasion, after passing the peak of clinical symptoms, has the potential to continue in various other parts of the body, showing symptoms like pain in the muscles [45,46,52,53].

Therefore, patients recovering from COVID-19 often experience persistent issues such as muscle pain. A therapeutic agent like sACE2-Anti-CD16VHH could potentially alleviate these symptoms.

In addition to its therapeutic potential, sACE2-AntiCD16VHH may be utilized to prevent COVID-19 infections. Furthermore, it can be used as a therapeutic drug in future outbreaks of any new coronavirus that uses the ACE2 receptor. Thus, with the end of the COVID-19 pandemic, its date of use will not expire.

If SARS-CoV-2 escapes ACE2 neutralization by decreasing its affinity, the virus could mutate into a less pathogenic virus, similar to the re-emergent SARS-CoV in 2003–2004, having a lower affinity for ACE2, resulting in less severe infection/no secondary transmission [48,50]. Thus, SARS-CoV-2 encounters an evolutionary trap when faced with sACE2-anti-CD16VHH therapy, leading toward a more benign clinical course.

Studies indicate that the level of IgG in COVID-19 patients with severe symptoms is higher than in patients with mild symptoms. It has been hypothesized that at the onset of the disease, the immune system of severely affected patients does not stop the virus at the onset of the infection due to an impairment of the IFN-I response. Perhaps it may multiply in the body tissues widely and lead to severe clinical symptoms [54]. The immune system may then produce a burst of antibodies, and these IgG antibodies may cause more inflammation, most likely through complement activation and leukocyte recruitment, than protection. As the AntiCD16VHH of the sACE2-AntiCD16VHH fusion protein, with its high affinity for the NK cell’s Fcγ receptor, can efficiently link the NK cells to the SARS-CoV-2-infected cells, the novel sACE2-AntiCD16VHH fusion protein could be a good immunotherapeutic agent to stop the spread of infection at the onset of the disease.

## 5. Conclusions

The findings show that a VHH directed against CD16 could be an excellent candidate to be linked to soluble ACE2 to generate a bi-specific molecule (sACE2-AntiCD16VHH) suitable for bridging effector cells and infected target cells to inhibit SARS-CoV-2 variant spike proteins and pro-inflammatory cytokines and to activate natural killer cell cytotoxicity against virus-infected cells. It is warranted to investigate this further in future studies that test this approach in preclinical animal studies.

## Figures and Tables

**Figure 1 vaccines-13-00199-f001:**
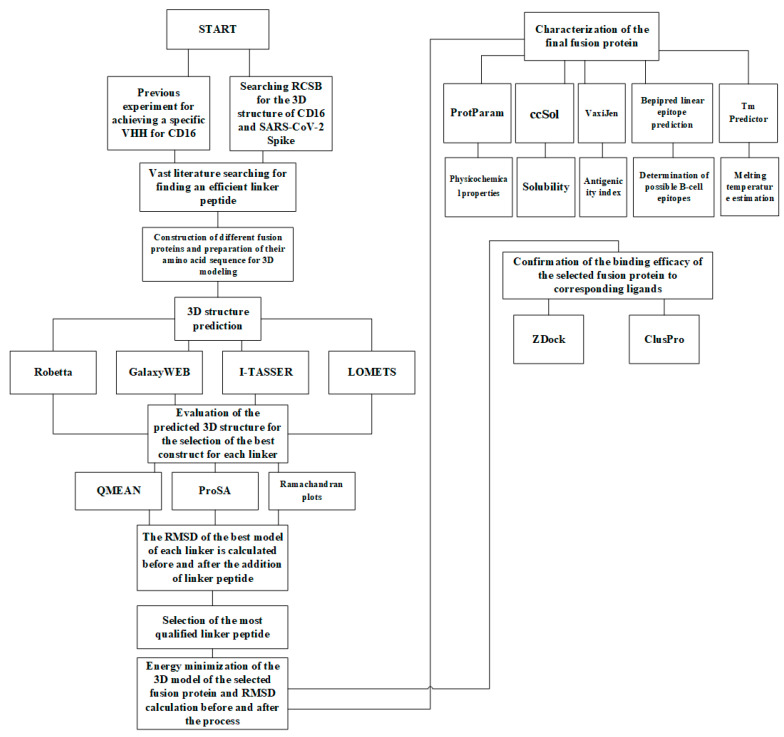
A detailed workflow of the whole sACE2-anti-CD16VHH design experiment.

**Figure 2 vaccines-13-00199-f002:**
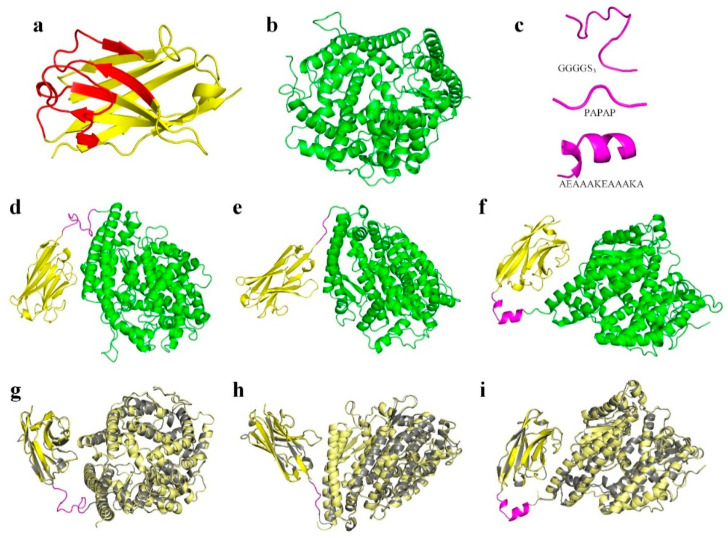
The 3D structures of the CD16-specific VHH, soluble ACE2, linker peptides, VHH-GGGGS_3_-ACE2, VHH-PAPAP-ACE2, and VHH-AEAAAKEAAAKA-ACE2 and the structural alignment of the fusion constructs with the VHH and soluble ACE2. (**a**) The 3D model of the CD16-specific VHH with the framework regions presented in yellow and the complementarity-determining regions presented in red. (**b**) The 3D structure of the soluble ACE2 (aa 18–615). (**c**) The cartoon presentation of the GGGGS_3_, PAPAP, and AEAAAKEAAAKA linker peptides. (**d**–**f**) The predicted 3D structures of VHH-GGGGS_3_-ACE2, VHH-PAPAP-ACE2, and VHH-AEAAAKEAAAKA-ACE2, respectively. The VHH is presented in yellow, the linker peptide in magenta, and ACE2 in green. (**g**–**i**) The structural alignment of VHH-GGGGS_3_-ACE2, VHH-PAPAP-ACE2, and VHH-AEAAAKEAAAKA-ACE2, respectively, with the CD16-specific VHH and ACE2. The fusion construct is presented in gray, with the linker peptide in magenta, the monomeric VHH in yellow, and the soluble ACE2 in pale yellow.

**Figure 3 vaccines-13-00199-f003:**
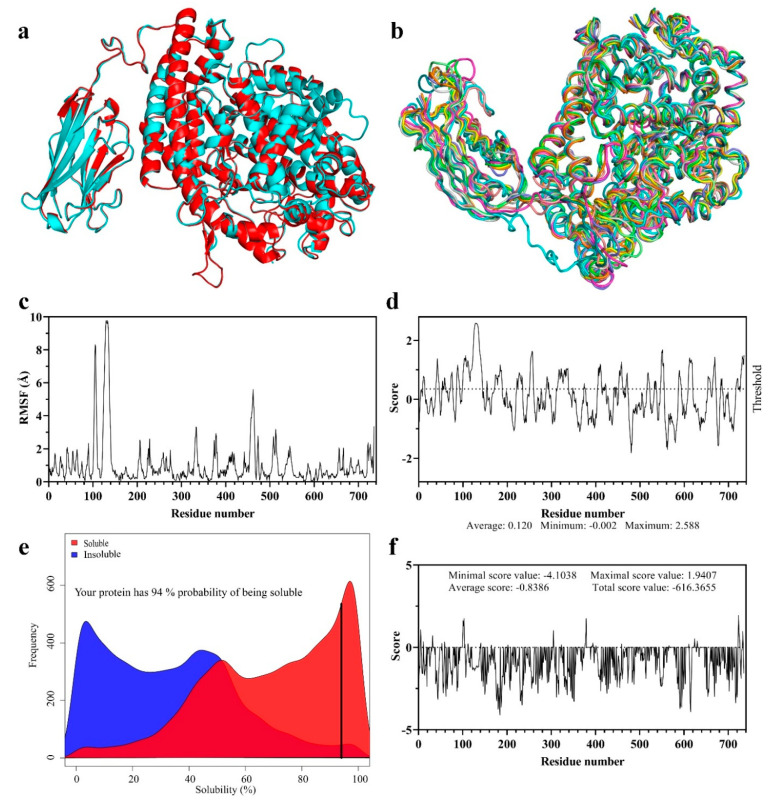
Energy minimization, flexibility assessment, antigenicity, and the solubility profile of VHH-GGGGS_3_-ACE2. (**a**) The structural alignment of the energy-minimized VHH-GGGGS_3_-ACE2 with its native counterpart. The structure depicted in cyan shows VHH-GGGGS_3_-ACE2 before energy minimization, and the structure in red shows VHH-GGGGS3-ACE2 after energy minimization. (**b**) Evaluation of the flexibility of VHH-GGGGS_3_-ACE2 by the CABS-flex 2.0 server. Ten structures predicted during the flexibility are presented in different colors (in the ribbon presentation). (**c**) The RMSF plot of VHH-GGGGS_3_-ACE2. (**d**) The Bepipred Linear Epitope Prediction results with the threshold set at 0.350. (**e**) The solubility propensity of VHH-GGGGS_3_-ACE2, as assessed by ccSol omics. (**f**) The solubility profile of VHH-GGGGS_3_-ACE2 as predicted by the Aggrescan3D server.

**Figure 4 vaccines-13-00199-f004:**
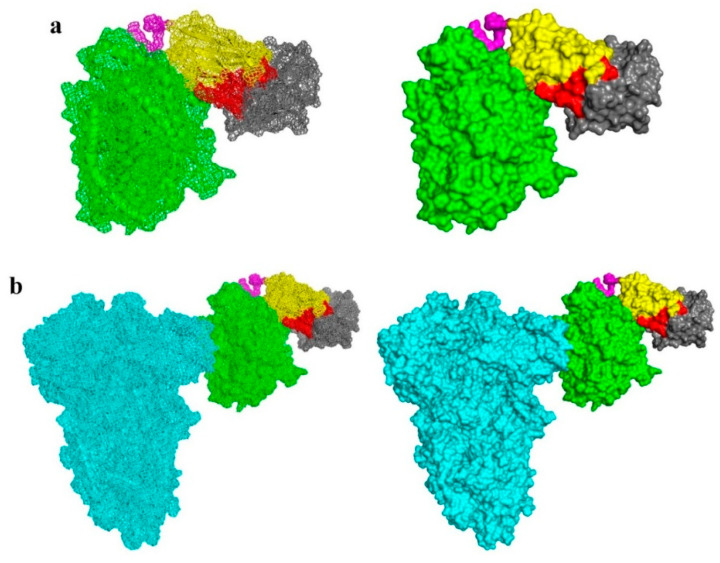
The docking of VHH-GGGGS_3_-ACE2 to CD16 and SARS-CoV-2 Spike. (**a**) VHH-GGGGS_3_-ACE2 as docked to CD16 in mesh and surface presentation (left and right, respectively). (**b**) VHH-GGGGS_3_-ACE2 as docked to CD16 and SARS-CoV-2 Spike in mesh and surface presentation (left and right, respectively). The VHH framework regions are presented in yellow, the complementarity-determining regions in red, the linker peptide in magenta, ACE2 in green, CD16 in gray, and SARS-CoV-2 Spike in cyan.

**Figure 5 vaccines-13-00199-f005:**
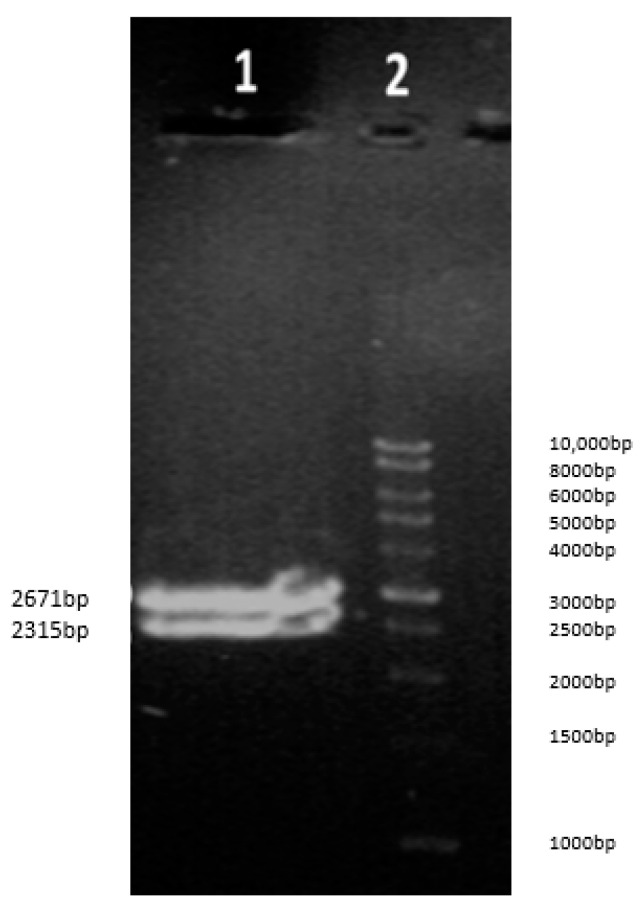
Enzymatic digestion of soluble pUC57/ACE2-AntiCD16VHH by *EcoRI* and *BamHI* (Fermentase). (**1**) Digestion products, (**2**) DNA Ladder. The digestion protocol involved mixing 1 μg of plasmid DNA with 5 μL of 10× buffer, 10 units of the respective restriction enzyme (*Eco*RI and *Bam*HI), and nuclease-free water to a final volume of 50 μL. The mixture reagent was incubated at 37 °C for 1 h. The digested DNA was then separated by 1% agarose gel.

**Figure 6 vaccines-13-00199-f006:**
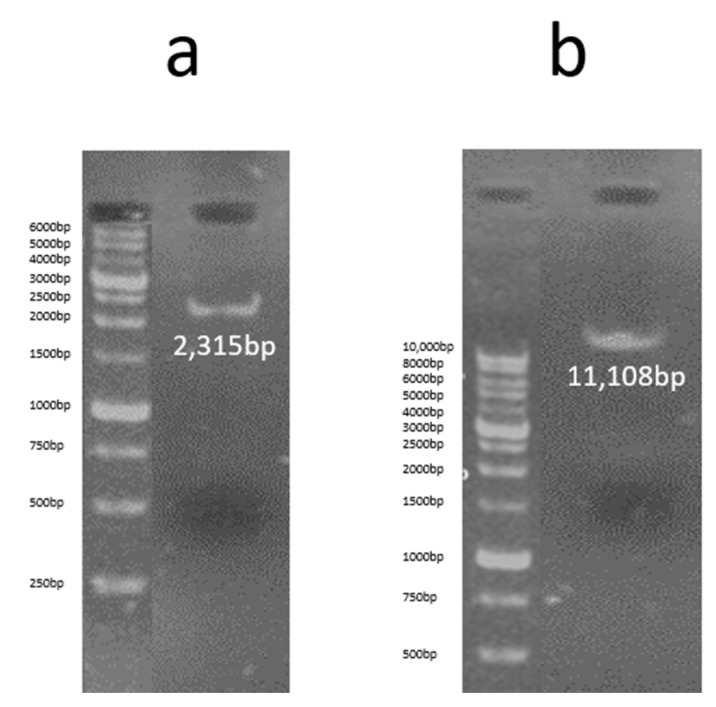
(**a**) The gel purification product related to sACE2-AntiCD16VHH fragment. Following enzymatic digestion of the soluble pUC57/ACE2-AntiCD16VHH molecule using *Eco*RI and *Bam*HI, the 2315 bp band corresponding to the sACE2-AntiCD16VHH gene fragment was further purified using a gel extraction kit (Roche), as per the kit’s instructions. The purified fragment was then electrophoresed on a 1% *w*/*v* agarose gel. (**b**) The cleaned up product of pLVX-EF1alpha-IRES-Puro vector after enzymatic digestion with *EcoRI* and *BamHI* enzymes. The pLVX-EF1alpha-IRES-Puro vector was digested with *Eco*RI and *Bam*HI enzymes, and the resulting fragment (11,108 bp) was purified using the Roche Clean-up kit according to the kit’s instructions. The purified fragment was then electrophoresed on a 1% *w*/*v* agarose gel.

**Figure 7 vaccines-13-00199-f007:**
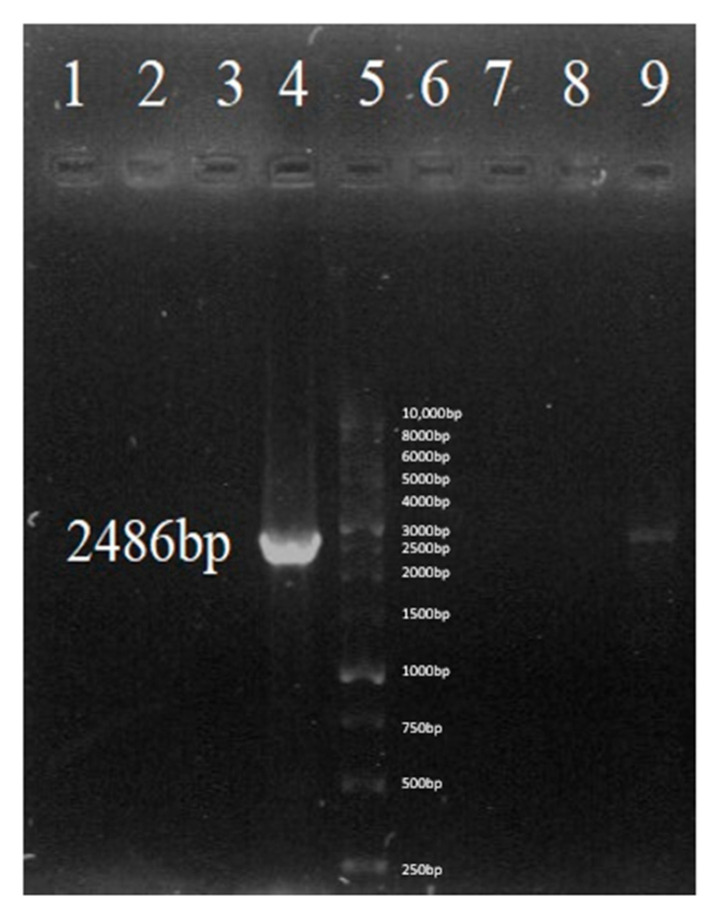
Confirmation of sACE2-AntiCD16VHH cloning in pLVX-EF1alpha-IRES-Puro vector via colony PCRs 1, 2, 3, 4, 6, 7, 8, and 9. Colony PCR products of 8 different colonies. 5. DNA Ladder. Colony PCR was conducted to verify the successful cloning of the sACE2-AntiCD16VHH gene into the pLVX-EF1alpha-IRES-Puro vector. Following transformation, colony PCRs were performed on 8 different colonies using forward and reverse primers specifically designed to target regions just before and after the sACE2-AntiCD16VHH gene insertion site within the pLVX plasmid. These primers amplify a fragment of 2486 bp, confirming the presence and correct integration of the sACE2-AntiCD16VHH gene. PCR products were electrophoresed on a 1% *w*/*v* agarose gel.

**Figure 8 vaccines-13-00199-f008:**
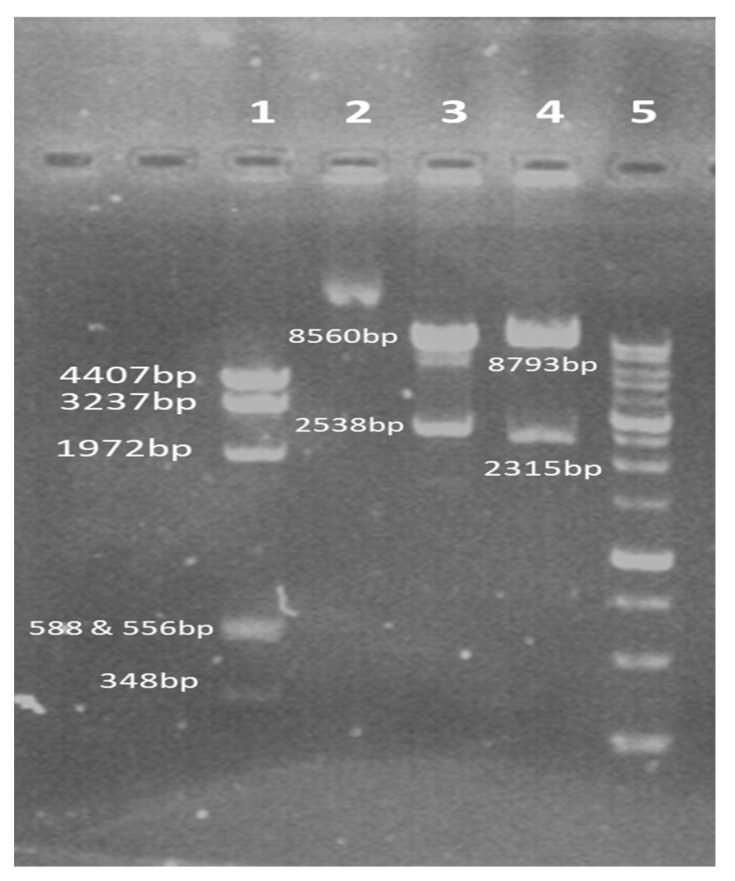
The result of enzymatic digestion of recombinant pLVX/sACE2-AntiCD16VHH. 1. Digestion by *HindIII*. 2. pLVX/sACE2-AntiCD16VHH (undigested). 3. Digestion by *XhoI.* 4. Simultaneous digestion with *EcoRI* and *BamHI*. 5. DNA Ladder.

**Figure 9 vaccines-13-00199-f009:**
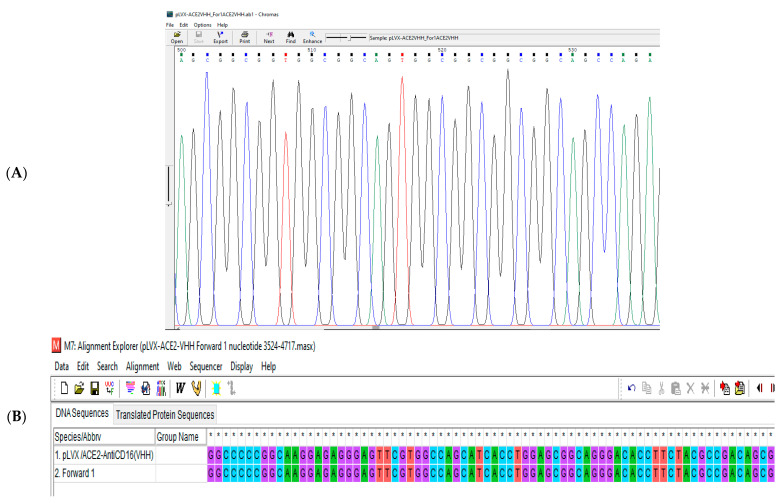
(**A**) A part of the sequencing result of pLVX/sACE2-AntiCD16VHH using forward primer 1. (**B**) A part of the sequence alignment of the pLVX/sACE2-AntiCD16VHH using forward primer 1. The result showed that nucleotides 3524 to 4717 are completely correct and without any mutations. * GGCC are nuclei symbols.

**Figure 10 vaccines-13-00199-f010:**
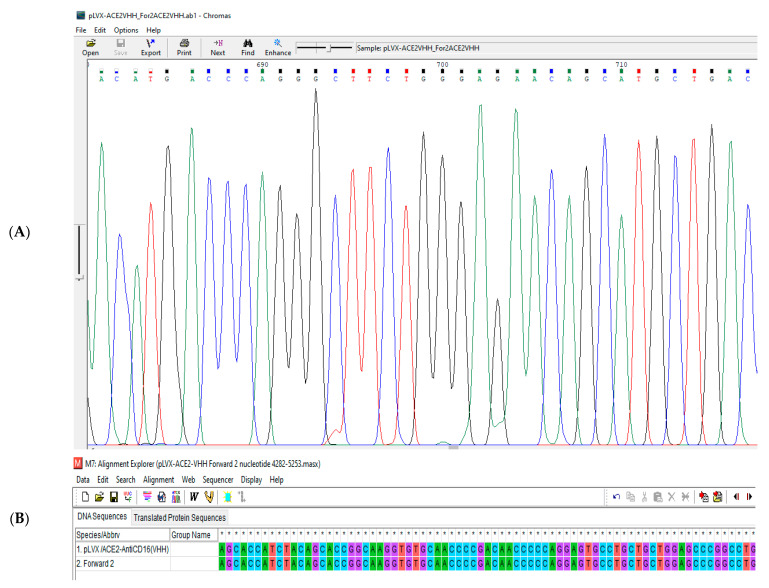
(**A**) A part of the sequencing result of pLVX/ACE2-AntiCD16(VHH) using forward primer 2. (**B**) A part of the sequence alignment of the pLVX/ACE2-AntiCD16(VHH) using forward primer 2. The result showed that nucleotides 4282 to 5282 are completely correct and without any mutations. * GGCC are nuclei symbols.

**Figure 11 vaccines-13-00199-f011:**
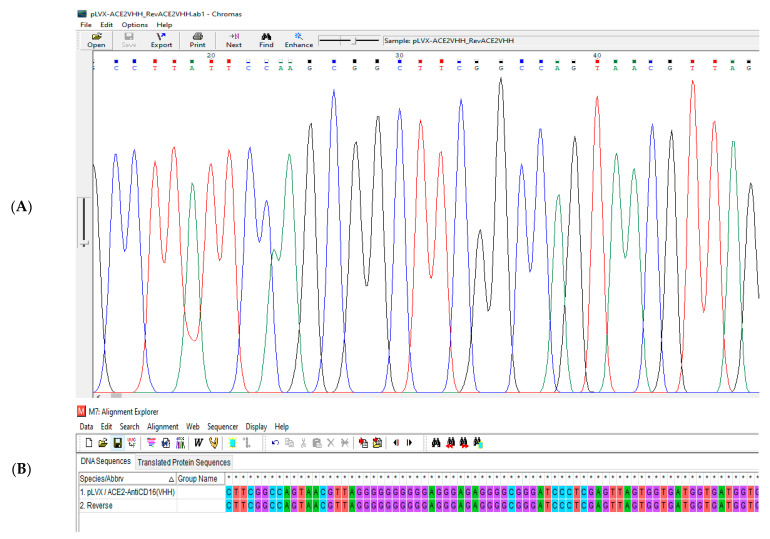
(**A**) A part of the sequencing result of pLVX/ACE2-AntiCD16(VHH) using the reverse primer. (**B**) A part of the sequence alignment of the pLVX/ACE2-AntiCD16(VHH) using the reverse primer. The result showed that nucleotides 5814 to 5917 are completely correct and without any mutations. * GGCC are nuclei symbols.

**Figure 12 vaccines-13-00199-f012:**
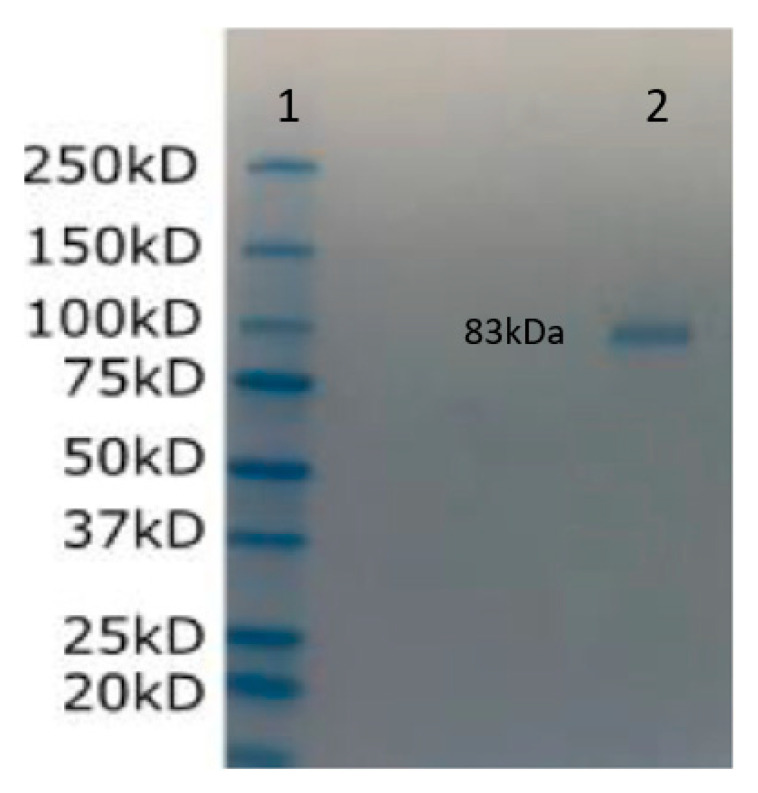
SDS-PAGE (Sodium Dodecyl Sulfate–Polyacrylamide Gel Electrophoresis) analysis of sACE2-CD16VHH. Lane 1, protein molecular weight marker; lane 2, sACE2-CD16VHH (MW: 83 kDa). The sample was applied to SDS–PAGE gel. The separation was performed on a polyacrylamide 12% gel at a constant voltage of 100 for 2 h. Then, the gel was stained with Coomassie blue.

**Figure 13 vaccines-13-00199-f013:**
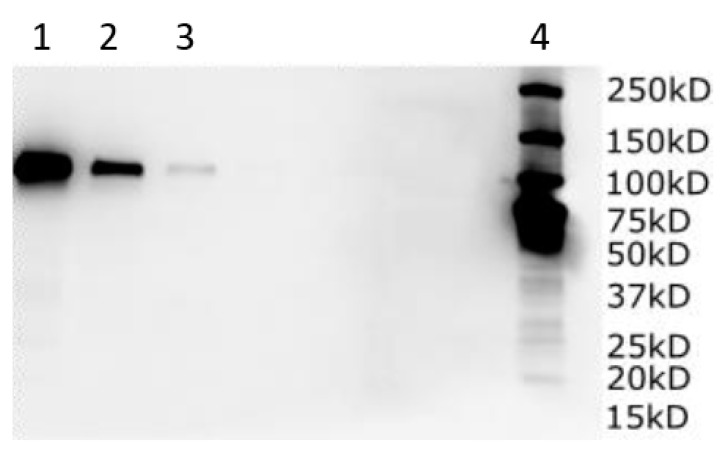
Western blot result of the sACE2-antiCD16VHH fusion protein with a His tag, using an anti-His tag antibody. Three samples were prepared and examined, each representing a different dilution factor. The samples were denatured in sample buffer and boiled at 95 °C for 5 min. Different concentrations of the sample, including undiluted (lane 1), 5-fold dilution (lane 2), and 25-fold dilution (lane 3), and protein molecular weight marker (lane 4), were resolved on an SDS-PAGE 12% gel and analyzed on a membrane using a wet transfer system at 100 V for 2 h. The PVDF membrane was treated with 5% BSA in TBST blocking solution for 1 h, then an anti-His tag antibody overnight at 4 °C. Chemiluminescent detection was performed using an ECL reagent and a CCD camera-based imager.

**Figure 14 vaccines-13-00199-f014:**
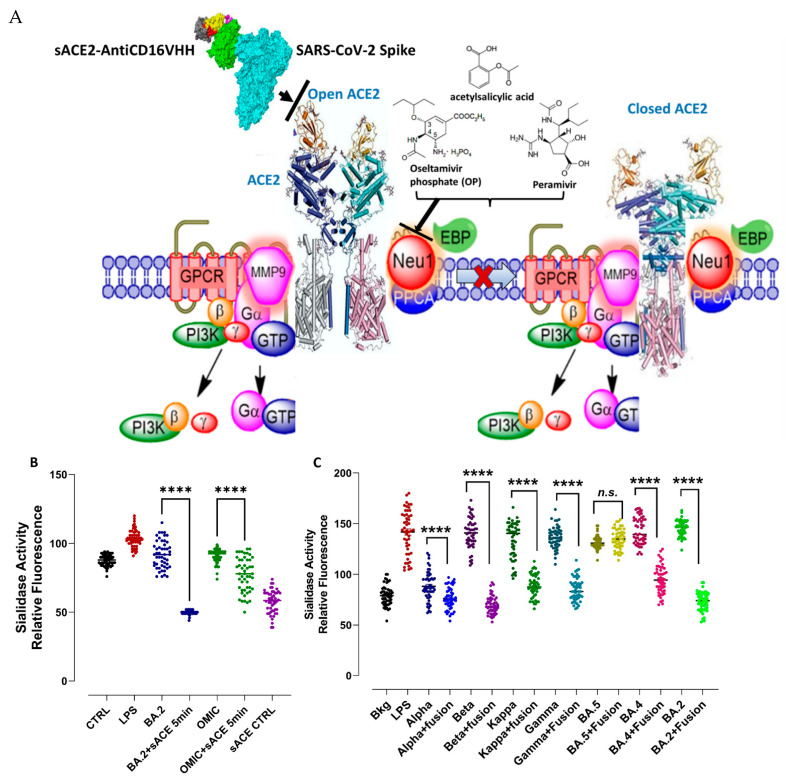
(**A**) The ACE2 receptor is proposed to exist in a multimeric receptor complex with neuromedin-B GPCR receptor, Neu1, and ACE2, in naïve (unstimulated) and stimulated ACE2-expressing cells. Here, a novel molecular signaling platform regulating the signaling and interaction signaling mechanism(s) between these molecules on the cell surface uncovers an S protein-induced ACE2 activation signaling axis mediated by Neu1 sialidase activation and the modification of ACE2 PD glycosylation. **Notes**: SARS-CoV-2 S proteins trigger an ACE2 conformational change to potentiate the NMBR-ACE2-MMP-9 signaling platform to activate Neu1 sialidase. Activated MMP-9 removes the EBP (elastin-binding protein) as part of the β-galactosidase protective protein cathepsin A (PPCA). Neu1 hydrolyzes α-2,3 sialyl residues at the ectodomain of ACE2 to remove steric hindrance to facilitate ACE2 subunit association, activation, and subsequent cellular entry. Oseltamivir phosphate (OP) and acetylsalicylic acid inhibit Neu1 sialidase activity. **Citation**: Taken partly from Harless et al. [41] Cells 2023, 12, 1332. https://doi.org/10.3390/cells12091332. Copyright: © 2023 by the authors. Licensee MDPI, Basel, Switzerland. This article is an open-access article distributed under the terms and conditions of the Creative Commons Attribution (CC BY) license (https://creativecommons.org/licenses/by/4.0/). (**B**,**C**) RAW-Blue macrophage cells were allowed to adhere to circular 12 mm glass slides for 24 h at 37 °C in a humidified incubator. After removing the media, 0.318 mM 4-MUNANA substrate in Tris-buffered saline, pH 7.4, was added to cells alone (background, Bkg) or with indicated S-proteins, 20 μg/mL or endotoxin LPS, 50 μg/mL. In combination with the indicated S-proteins and sACE2-AntiCD16VHH fusion protein, 200 μg/mL, the sialidase was markedly reduced. Fluorescent images were taken at 2 min after adding substrate using epi-fluorescent microscopy (40× objective). The mean fluorescence surrounding the live cells was measured using Image J software. A scatter plot of data point visualization represents fluorescence values (n = 50) from one experiment out of 3 independent experiments with similar results. The fluorescence values of each group were compared to the indicated group by ANOVA using Fisher’s LSD test with 95% confidence, with asterisks indicating statistical significance. Data represent the mean ± SEM of 3 independent experiments performed in triplicate displaying similar results. As indicated by asterisks, statistical significance was calculated with ANOVA and Fisher’s uncorrected LSD post hoc test at a confidence level of 95%. n.s. = non-significant, **** *p* < 0.0001.

**Figure 15 vaccines-13-00199-f015:**
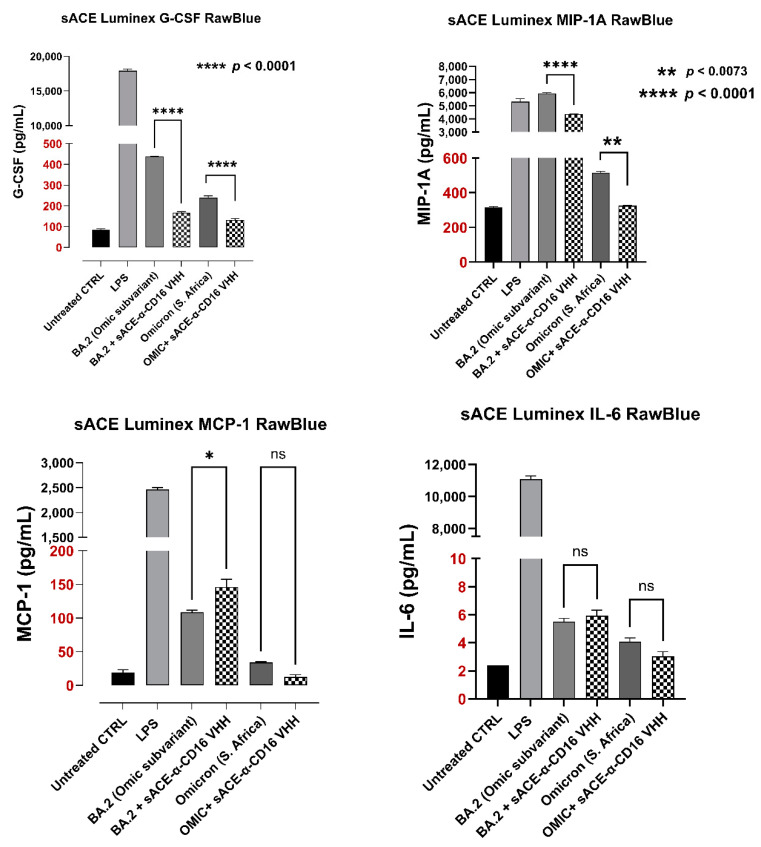
Effect of sACE2-AntiCD16VHH fusion protein on the inflammatory cytokine response induced by the Omicron and BA.2 variants of SARS-CoV-2 spike S proteins using the MILLIPLEX^®^ Luminex^®^ xMAP^®^ cytokine assay. Data represent the mean ± SEM of 3 separate, independent experiments performed in triplicate, displaying similar results. As indicated by asterisks, statistical significance was calculated with ANOVA and Fisher’s uncorrected LSD post hoc test at a confidence level of 95%. ns = non-significant, **** *p* < 0.0001, ** *p* < 0.01, * *p* < 0.05.

**Figure 16 vaccines-13-00199-f016:**
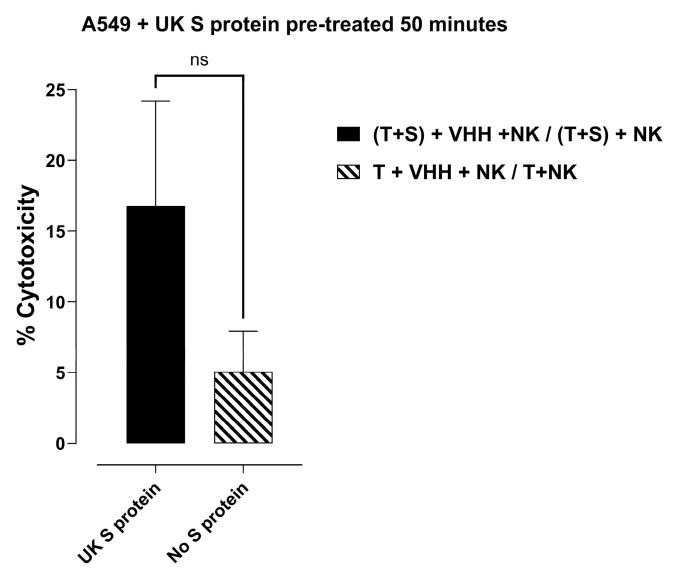
MACS separated human NK cells and ACE-2^+^ A549 target cells were co-cultured with sACE2-AntiCD16VHH fusion protein with or without (control) SARS-CoV-2 S protein (UK variant), and the NK cytotoxicity was measured by Alamar Blue colorimetric assay after 4 h. T, A549 cell; S, UK S-protein (alpha); VHH, sACE2-α-CD16 (3.4 μg/mL). Data represent the mean ± SEM of 2 separate, independent experiments performed in duplicate. As indicated, statistical significance was calculated with ANOVA and Fisher’s uncorrected LSD post hoc test at a confidence level of 95%. ns = non-significant, (*p* = 0.1389).

**Figure 17 vaccines-13-00199-f017:**
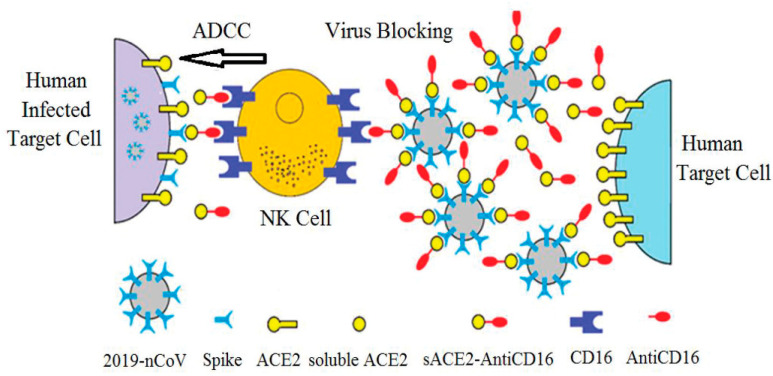
The coronavirus spike protein binds to ACE2 on target cells, leading to cell entry. The sACE2-AntiCD16VHH not only blocks SARS-CoV-2 from infecting cells but also mediates ADCC by NK cells.

**Table 1 vaccines-13-00199-t001:** Quality assessment of the best predicted 3D model of each server through assessing their QMEAN scores, ProSA Z-scores, and the Ramachandran plots.

Construct Designation	Server Used for Structure Prediction	QMEANDisCo (Global Score)	ProSa (Z-Score)	Ramachandran Analysis			
				Residues in Favored Regions (>98%)	Residues in Allowed Regions (>99.8%)	Outliers (<0.05%)	Rama Distribution Z-Score (abs(Z-Score) < 2)
VHH-GGGGS1-ACE2	Robetta Model 1	0.84 ± 0.05	−14.32	97.6% (706/723)	99.7% (721/723)	2 (0.28%)	0.35 ± 0.29
	GalaxyWEB Model 1	0.84 ± 0.05	−14.22	98.3% (711/723)	99.6% (720/723)	3 (0.41%)	−1.37 ± 0.27
	I-TASSER Model 1	0.68 ± 0.05	−8.79	79.4% (574/723)	89.5% (647/723)	76 (10.51%)	−4.43 ± 0.26
	LOMETS Model 1	0.68 ± 0.05	−11.21	90.9% (657/723)	96.4% (697/723)	26 (3.60%)	1.26 ± 0.31
VHH-GGGGS2-ACE2	Robetta Model 1	0.84 ± 0.05	−14.8	98.1% (714/728)	99.5% (724/728)	4 (0.55%)	0.51 ± 0.30
	GalaxyWEB Model 1	0.83 ± 0.05	−14.35	98.2% (715/728)	99.5% (724/728)	4 (−1.24 ± 0.26)	−1.24 ± 0.26
	I-TASSER Model 1	0.64 ± 0.05	−8.4	77.1% (561/728)	88.7% (646/728)	82 (11.26%)	−4.67 ± 0.25
	LOMETS Model 1	0.68 ± 0.05	−11.89	90.4% (658/728)	96.6% (703/728)	25 (3.43%)	1.35 ± 0.31
VHH-GGGGS3-ACE2	Robetta Model 1	0.83 ± 0.05	−14.47	98.1% (719/733)	99.9% (732/733)	1 (0.14%)	0.39 ± 0.29
	GalaxyWEB Model 1	0.84 ± 0.05	−14.52	98.6% (723/733)	99.7% (731/733)	2 (0.27%)	−1.10 ± 0.27
	I-TASSER Model 1	0.65 ± 0.05	−7.99	75.4% (553/733)	88.3% (647/733)	86 (11.73%)	−4.54 ± 0.25
	LOMETS Model 1	0.73 ± 0.05	−12.77	90.5% (663/733)	96.9% (710/733)	23 (3.14%)	1.66 ± 0.31
ACE2-GGGGS1-VHH	Robetta Model 1	0.78 ± 0.05	−12.36	97.4% (704/723)	99.6% (720/723)	3 (0.41%)	0.47 ± 0.30
	GalaxyWEB Model 1	0.84 ± 0.05	−14.12	97.9% (708/723)	99.6% (720/723)	3 (0.41%)	−1.40 ± 0.26
	I-TASSER Model 1	0.71 ± 0.05	−9.58	78.2% (526/673)	91.1% (613/673)	60 (8.92%)	−5.09 ± 0.24
	LOMETS Model 1	0.74 ± 0.05	−13.15	92.4% (668/723)	97.9% (708/723)	15 (2.07%)	1.68 ± 0.30
ACE2-GGGGS2-VHH	Robetta Model 1	0.84 ± 0.05	−13.74	98.4% (716/728)	99.9% (727/728)	1 (0.14%)	0.79 ± 0.30
	GalaxyWEB Model 1	0.84 ± 0.05	−14.15	97.3% (708/728)	99.5% (724/728)	4 (0.55%)	−1.23 ± 0.27
	I-TASSER Model 1	0.70 ± 0.05	−10.09	79.9% (582/728)	91.5% (666/728)	62 (8.52%)	−4.24 ± 0.27
	LOMETS Model 1	0.71 ± 0.05	−12.9	86.8% (632/728)	94.8% (690/728)	38 (5.22%)	1.23 ± 0.31
ACE2-GGGGS3-VHH	Robetta Model 1	0.78 ± 0.05	−12.6	98.1% (719/733)	99.6% (730/733)	3 (0.41%)	0.58 ± 0.29
	GalaxyWEB Model 1	0.84 ± 0.05	−14.09	97.7% (716/733)	99.5% (729/733)	4 (0.55%)	−1.40 ± 0.26
	I-TASSER Model 1	0.70 ± 0.05	−10.23	76.7% (562/733)	90.3% (662/733)	71 (9.69%)	−4.80 ± 0.25
	LOMETS Model 1	0.74 ± 0.05	−13.37	92.8% (680/733)	98.0% (718/733)	15 (2.05%)	2.00 ± 0.30
VHH-PAPAP-ACE2	Robetta Model 1	0.83 ± 0.05	−14.51	98.1% (709/723)	99.6% (720/723)	3 (0.41%)	0.66 ± 0.30
	GalaxyWEB Model 1	0.83 ± 0.05	−14.41	98.1% (709/723)	99.7% (721/723)	2 (0.28%)	−0.95 ± 0.26
	I-TASSER Model 1	0.69 ± 0.05	−12.86	81.2% (587/723)	92.5% (669/723)	54 (7.47%)	−3.91 ± 0.26
	LOMETS Model 1	0.71 ± 0.05	−13.26	92.1% (666/723)	97.5% (705/723)	18 (2.49%)	1.54 ± 0.30
ACE2-PAPAP-VHH	Robetta Model 1	0.77 ± 0.05	−12.3	97.8% (707/723)	99.9% (722/723)	1 (0.14%)	0.78 ± 0.30
	GalaxyWEB Model 1	0.83 ± 0.05	−13.69	98.3% (711/723)	99.7% (721/723)	2 (0.28%)	−1.09 ± 0.27
	I-TASSER Model 1	0.70 ± 0.05	−9.56	80.5% (582/723)	92.4% (668/723)	55 (7.61%)	−4.68 ± 0.24
	LOMETS Model 1	0.72 ± 0.05	−13.16	89.1% (644/723)	96.0% (694/723)	29 (4.01%)	1.68 ± 0.31
VHH-AEAAAKEAAAKA-ACE2	Robetta Model 1	0.82 ± 0.05	−13.96	97.8% (714/730)	99.7% (728/730)	2 (0.27%)	0.81 ± 0.30
	GalaxyWEB Model 1	0.82 ± 0.05	−13.81	97.8% (714/730)	100.0% (730/730)	0 (0.00%)	−1.43 ± 0.26
	I-TASSER Model 1	0.66 ± 0.05	−8.26	79.3% (579/730)	89.6% (654/730)	76 (10.41%)	−4.37 ± 0.25
	LOMETS Model 1	0.71 ± 0.05	−12.74	92.5% (675/730)	97.9% (715/730)	15 (2.05%)	1.72 ± 0.30
ACE2-AEAAAKEAAAKA-VHH	Robetta Model 1	0.76 ± 0.05	−12.34	97.1% (709/730)	99.5% (726/730)	4 (0.55%)	0.86 ± 0.29
	GalaxyWEB Model 1	0.82 ± 0.05	−13.76	98.2% (717/730)	99.6% (727/730)	3 (0.41%)	−1.10 ± 0.27
	I-TASSER Model 1	0.67 ± 0.05	−9.64	80.0% (584/730)	89.5% (653/730)	77 (10.55%)	−3.94 ± 0.26
	LOMETS Model 1	0.71 ± 0.05	−13.01	91.8% (670/730)	96.6% (705/730)	25 (3.42%)	1.83 ± 0.30

## Data Availability

All data needed to evaluate the conclusions in the paper are present.

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
