# Peer review of "Novel sACE2-Anti-CD16VHH Fusion Protein Surreptitiously Inhibits SARS-CoV-2 Variant Spike Proteins and Macrophage Cytokines, and Activates Natural Killer Cell Cytotoxicity"

_vaccines, 2025, doi:10.3390/vaccines13020199_

Round 1

Reviewer 1 Report

Comments and Suggestions for Authors

Thanks for submitting this paper.

Please reference your prior work and correct the statement on line 92.

"  we generated the sACE2-AntiCD16VHH fusion protein for the first time."

Seems you are extending your prior work in PMID 32663051 and is not the first time.

The legend of Fig 14 is incomplete. 

The main result of the effect of sACE2-An- 658 tiCD16VHH on cytotoxicity in Fig 17 does not seem to be significant.  Please tone down the conclusions to reflect that or add a statistical proof of effectiveness. The error bars (SD??) need to be explained and documented as significant or not.

Author Response

Reviewer #1:
Comments and Suggestions for Authors
Thanks for submitting this paper.

Please reference your prior work and correct the statement on line 92.

"  we generated the sACE2-AntiCD16VHH fusion protein for the first time."

Seems you are extending your prior work in PMID 32663051 and is not the first time.
Author Response: Thanks for your comment. The PMID 32663051 article was just a commentary paper in which we suggested to generate the sACE2-AntiCD16VHH fusion protein. We have corrected the sentence and made the citation: Ref: Sheikhi A, Hojjat-Farsangi M. An immunotherapeutic method for COVID-19 patients: a soluble ACE2-Anti-CD16 VHH to block SARS-CoV-2 Spike protein. Hum Vaccin Immunother. 2020 Jul 14:1-6. doi: 10.1080/21645515.2020. 1787066. 

The legend of Fig 14 is incomplete.

Answer: Thanks for your comment. We have corrected the figure by adding text in the legend.

The main result of the effect of sACE2-AntiCD16VHH on cytotoxicity in Fig 17 does not seem to be significant.  Please tone down the conclusions to reflect that or add a statistical proof of effectiveness. The error bars (SD??) need to be explained and documented as significant or not.

Answer: Thanks for your comment. The stats used are in the figure legend. We have corrected the text to indicate that the results show a marked increase in cytotoxicity, but it is non-significant.

Reviewer 2 Report

Comments and Suggestions for Authors

The present manuscript entitled “Novel sACE2-Anti-CD16VHH Fusion Protein Surreptitiously Inhibits SARS-CoV-2 Variant Spike Proteins, Macrophage Cytokines and Activates Natural Killer Cell Cytotoxicity” by Sheikhi describes the efficiency of newly developed fusion protein sACE2-Anti-CD16VHH in combating SARS-CoV-2 infections by multiple effects. 1) blocking the virus's entry into cells by targeting its spike protein. 2) Reducing harmful inflammation by suppressing macrophage cytokine production. 3) Boosting the immune response by activating NK cell's cytotoxicity to kill infected cells.

This study provides valuable insights into SARS-COV-2 infection and its potential therapeutic interventions. Although the study is well-designed and executed, several limitations need to be addressed. Below are my comments-

Major comments-

1.     The authors checked the efficiency of fusion protein on enhanced NK cell cytotoxicity. However, the authors have not checked the potential of this fusion protein in vivo model.  

2.    The authors described this fusion protein reduced the proinflammatory cytokines by macrophages. But it was not checked it is because of some side effects on the macrophages like cell death caused by the fusion protein.  

3.    The authors described that fusion protein blocks SARS-COV2 S-protein binding to ACE2 however the authors did not check viral replication in the cells in the presence and absence of this fusion protein.  

4.    Type I IFN has been reported to function as both pathogenic and protective cytokines in case of COVID-19 infection. Authors should have checked the level of this cytokine in the macrophage also along with other pro-inflammatory cytokines checked by them.

Minor comments-

1.     Line 91: What is AntiCD16VHH? Please explain. Why did the authors use this protein? What were the advantages?

2.    Lines 90-93: Please add a few more lines to show what was done with this protein and what was the outcome.

3.    Lines 235-239: What was the NK cells yield? Explain the methods in brief. Did the authors check the purity?

4.    Figures 6 A and B: DNA ladder as a marker is missing. Please add that.

5.    Figure 7: The band 2486 bp appears larger than 2600 bp if compared to the DNA ladder.

6.    Lines 427-428: “construct is completely correct” should be “construct was accurate”.

7.    Lines 501-502: What was the yield of fusion protein?

8.    Lines 519-520: “Making use of an anti-His tag antibody” should be “by using anti-His antibody”.

9.    Line 542: Please abbreviate ARDS.

10.                    Line 607: Please abbreviate ADCC

11.  Lines 605-610: These lines appear like material and methods. Please discuss the results.

12. Figure 17: The formula on the right side of the image should be discussed in materials and methods and not here. Please remove it from the figure.

13. Lines 617-618 are grammatically incorrect. Please fix it.

14. Lines 628-629: “ The Fc fragment of the chimeric molecule has a longer half-life on sACE2” should be “The Fc fragment of the chimeric molecule extends the half-life of sACE2”.

15. Lines 710-12: these lines have several errors. It should be “Patients recovering from COVID-19 often experience persistent issues such as muscle pain. A therapeutic agent like sACE2-Anti-CD16VHH could potentially alleviate these symptoms”.

16. Line 716: This line is “ Thus, with the end of the COVID-19 pandemic, its date of use won’t expire.” Is informal. Correct it.

17. Line 733: Change “medicine” to “agent”.

Author Response

This study provides valuable insights into SARS-COV-2 infection and its potential therapeutic interventions. Although the study is well-designed and executed, several limitations need to be addressed. Below are my comments-

Major comments-

1. The authors checked the efficiency of fusion protein on enhanced NK cell cytotoxicity. However, the authors have not checked the potential of this fusion protein in vivo model.

Answer: Thanks so much for your comment. As you suggested, one limitation of our study is that we did not check if the sACE2-AntiCD16VHH can augment NK cytotoxicity against SARS-CoV-2 transfected cells in vivo. We have added a sentence about the mentioned limitation to the discussion section. So, we revised the manuscript as you suggested.

2. The authors described this fusion protein reduced the proinflammatory cytokines by macrophages. But it was not checked it is because of some side effects on the macrophages like cell death caused by the fusion protein.

Answer: Thanks so much for your comment. As it is shown in Fig 16, the secreted cytokines by un-treated macrophages are almost equal to S proteins + sACE2-AntiCD16VHH Fusion protein treated macrophages which is obviously less than S protein treated macrophages. So, we can conclude that the cytokine reduction is due to the effect of the fusion protein. The fusion protein only did not have any adverse side effects on the cells based on the results of the sialidase assay.

  1. The authors described that fusion protein blocks SARS-COV2 S-protein binding to ACE2 however the authors did not check viral replication in the cells in the presence and absence of this fusion protein.

Answer: Thanks so much for your comment. As you suggested, one limitation of our study is that we did not check if the sACE2-AntiCD16VHH can block viral replication in the cells. We have added a sentence about the mentioned limitation to the discussion section. So, we revised the manuscript as you suggested. Also, we did not use the virus in the study but only the S proteins.

4. Type I IFN has been reported to function as both pathogenic and protective cytokines in case of COVID-19 infection. Authors should have checked the level of this cytokine in the macrophage also along with other pro-inflammatory cytokines checked by them.

Answer: Thanks so much for your comment. Although in some conditions, type 1 IFN has an inflammatory role, it has a protective role in the early stages of COVID-19 disease and, if secreted in sufficient quantities at the time of the coronavirus's entry into the body, can play a significant role in stopping the virus and activating NK cells. Therefore, due to its more important role in regulating the immune response, we did not consider it as a pro-inflammatory cytokine.

Minor comments-

1. Line 91: What is AntiCD16VHH? Please explain. Why did the authors use this protein? What were the advantages?

Answer: Thanks so much for your comment. All about the AntiCD16VHH has been explained in the introduction and discussion section on lines 644-659.

2. Lines 90-93: Please add a few more lines to show what was done with this protein and what was the outcome.

Answer: Thanks so much for your comment. We have added the below text to the introduction: The sACE2-AntiCD16VHH consists of the soluble ACE2 and the variable domain of the anti-Human CD16 nanobody (VHH) obtained from a llama, joined together by a linker. We showed the sACE2-AntiCD16VHH blocks different variants of SARS-CoV-2 spike molecules to bind to ACE-2 on the RawBlue macrophages. It also inhibited the secretion of pro-inflammatory cytokines from the SARS-CoV-2 S protein stimulated RawBlue macrophages and mediated ADCC by NK cells against SARS-CoV-2 S protein pretreated ACE2+ A549 target cells.

  1. Lines 235-239: What was the NK cells yield? Explain the methods in brief. Did the authors check the purity?

Answer: Thanks so much for your comment. NK purity and yield were checked in our previous study (Ref. no. 35). We have added the following in the text: Briefly, the whole blood sample was added to the required tube. The Isolation Cocktail was mixed to the sample and incubated at RT, 5 min. The RapidSpheres™ was mixed to sample. With RPMI, the sample volume was topped up to 2.5ml. Mixed by gently pipetting. The tube was placed into the magnet and incubated at RT, 3 min. The magnet was picked up, and in one continuous motion, the magnet and tube inverted, and the enriched NK cell suspension was poured into a new tube.

  1. Figures 6 A and B: DNA ladder as a marker is missing. Please add that.

Answer: Thanks so much for your comment. Done

  1. Figure 7: The band 2486 bp appears larger than 2600 bp if compared to the DNA ladder.

Answer: Thanks so much for your comment. The 2500 bp band is below the 3000 bp band and the 2000 bp band is after it, so the fusion protein band is located close to the 2500 bp band. The new figure shows the correct location of the aforementioned bands.

  1. Lines 427-428: “construct is completely correct” should be “construct was accurate”.

Answer: Thanks so much for your comment. We have corrected it.

7. Lines 501-502: What was the yield of fusion protein?

Answer: Thanks so much for your comment. We have added the following in the M&M: The fusion protein concentration was 643 µg/ml

8. Lines 519-520: “Making use of an anti-His tag antibody” should be “by using anti-His antibody”.

Answer: Thanks so much for your comment. DONE

9. Line 542: Please abbreviate ARDS.

Answer: Thanks so much for your comment. DONE

10. Line 607: Please abbreviate ADCC

Answer: DONE

11. Lines 605-610: These lines appear like material and methods. Please discuss the results.

Answer: Thanks so much for your comment. DONE

  1. Figure 17: The formula on the right side of the image should be discussed in materials and methods and not here. Please remove it from the figure.

Answer: Thanks so much for your comment. DONE

  1. Lines 617-618 are grammatically incorrect. Please fix it.

Answer: Thanks so much for your comment. FIXED

  1. Lines 628-629: “ The Fc fragment of the chimeric molecule has a longer half-life on sACE2” should be “The Fc fragment of the chimeric molecule extends the half-life of sACE2”.

Answer: Thanks so much for your comment. DONE

  1. Lines 710-12: these lines have several errors. It should be “Patients recovering from COVID-19 often experience persistent issues such as muscle pain. A therapeutic agent like sACE2-Anti-CD16VHH could potentially alleviate these symptoms”.

Answer: Thanks so much for your comment. DONE

  1. Line 716: This line is “ Thus, with the end of the COVID-19 pandemic, its date of use won’t expire.” Is informal. Correct it.

Answer: Thanks so much for your comment. DONE

  1. Line 733: Change “medicine” to “agent”.

Answer: Thanks so much for your comment. DONE

Round 2

Reviewer 2 Report

Comments and Suggestions for Authors

The authors have addressed most of my previous comments in the current revised version of the manuscript; however, the manuscript still needs improvement. Below are my suggestions to enhance the quality of the manuscript-

Line 41: “infected target cells to inhibit SARS-CoV-2 variant spike proteins” is confusing. It should be “infected target cells to inhibit SARS-CoV-2 variant spike proteins binding to ACE2 receptor on RAW cell line”.

Line 547: Catalog number is missing.

Line 603: The term “Separate” is not needed here.

Line 618: “secretion in response to omicron and BA.2” should be “secretion in response to omicron and BA.2 variants protein”.

Lines 620-621: “Notably, these inhibitory effects significantly reduced G-CSF and MIP-1A by approximately 99%”. In comparison with what? LPS? Please clarify it. It’s not clear.

Lines 695-697: The authors say that their generated fusion protein neutralizes the virus and prevents replication in the infected cells. However, they have not shown these data anywhere in the manuscript. They just checked if this fusion protein inhibits the binding of s protein to RAW cells. Please clarify this point.

Line 721: “transfected” should be “infected”.  

Line 751: Please correct the sentence. The sentence lacks fluency and could be rephrased for better clarity.

Figure 17 lacks important controls, such as NK cell alone, target cell alone, NK cell alone treated with fusion protein, target cell alone treated with fusion protein, and NK cell alone treated with s protein.

Lines 719-721: The authors have mentioned lack of in vivo as one of the limitations of this study, but they should have also explained why those aspects were not addressed.

Author Response

The authors have addressed most of my previous comments in the current revised version of the manuscript; however, the manuscript still needs improvement. Below are my suggestions to enhance the quality of the manuscript-

Author response: Thank you for the comments.

Line 41: “infected target cells to inhibit SARS-CoV-2 variant spike proteins” is confusing. It should be “infected target cells to inhibit SARS-CoV-2 variant spike proteins binding to ACE2 receptor on RAW cell line”.

Author response: Thank you for the comments. DONE

Line 547: Catalog number is missing. Catalog #: MAB050, Bio-Techne Canada

Line 603: The term “Separate” is not needed here. removed DONE

Line 618: “secretion in response to omicron and BA.2” should be “secretion in response to omicron and BA.2 variants protein”. DONE

Lines 620-621: “Notably, these inhibitory effects significantly reduced G-CSF and MIP-1A by approximately 99%”. In comparison with what? LPS? Please clarify it. It’s not clear. Compared to BA-2 subvariant alone - Added

Lines 695-697: The authors say that their generated fusion protein neutralizes the virus and prevents replication in the infected cells. However, they have not shown these data anywhere in the manuscript. They just checked if this fusion protein inhibits the binding of s protein to RAW cells. Please clarify this point. Therefore, based on data from this study, for the first time, we generated sACE2-antiCD16VHH fusion protein that has the potential to neutralize the virus and, at the same time, could prevent virus replication in cells.

Line 721: “transfected” should be “infected”.  DONE

Line 751: Please correct the sentence. The sentence lacks fluency and could be rephrased for better clarity. AntiCD16VHH may be utilized to prevent COVID-19 infections.

Figure 17 lacks important controls, such as NK cell alone, target cell alone, NK cell alone treated with fusion protein, target cell alone treated with fusion protein, and NK cell alone treated with s protein. The important controls are incorporated in the % cytotoxicity formula:

% cytotoxicity = [OD of S-pretreated target cells +VHH +NK / (OD of S-pretreated Target cells +NK)] x 100

% cytotoxicity = [(OD of untreated target cells +VHH +NK / (OD of Un-treated Target cells +NK)] x 100

Lines 719-721: The authors have mentioned lack of in vivo as one of the limitations of this study, but they should have also explained why those aspects were not addressed. 

One limitation of our study is that we did not check investigate if the sACE2-AntiCD16VHH can augment NK cytotoxicity against SARS-CoV-2 infected transfected cells in vivo due to biohazard limitations using viral infected cells in animals.